# DABstep: Data Agent Benchmark for Multi-step Reasoning

**Alex Egg**[1,♣,*] **Martin Iglesias Goyanes**[1,♣], **Friso Kingma**[1,♣], **Andreu Mora**[1,♣],
**Leandro von Werra**[2], **Thomas Wolf**[2]
[1]Adyen [2]Hugging Face
{alex.egg,martin.iglesiasgoyanes,friso.kingma,andreu.mora}@adyen.com

## Abstract

We introduce **DABstep**, a novel benchmark for evaluating AI agents on realistic
multi-step data analysis tasks. DABstep comprises over 450 real-world challenges
derived from a financial analytics platform, requiring models to combine code-
based data processing with contextual reasoning over heterogeneous documenta-
tion. Each task demands an iterative, multi-step problem-solving approach, testing
capabilities in data manipulation, cross-referencing multiple sources, and precise
result reporting. The benchmark provides a factoid-style answer format with au-
tomatic correctness checks for objective scoring at scale. We evaluate leading
LLM-based agents, revealing a substantial performance gap: even the best agent
achieves only 14.55% accuracy on the hardest tasks. We detail our benchmark's
design, dataset composition, task formulation, evaluation protocol, report baseline
results and analyze failure modes. DABstep is released with a public leaderboard
and toolkit to accelerate research in autonomous data analysis.

🌐 **Data & Code:** huggingface.co/spaces/adyen/DABstep
🤗 **Data & Dataset Card:** huggingface.co/datasets/adyen/dabstep

## 1 Introduction

Recent advances in large language models (LLMs) have enabled the development of autonomous
agentic workflows, particularly showing promise for automating complex, multi-step tasks within
domains like data science and software engineering. However, the evaluation of such agents, espe-
cially for data analysis, faces significant hurdles. Many existing benchmarks rely on synthetic tasks,
overly simplistic evaluations, or subjective assessment methods (such as LLM-as-a-judge, known for
biases), limiting their ability to accurately reflect the challenges encountered in real-world analytical
scenarios and gauge true agent capabilities.

To address these limitations, we introduce the Data Agent Benchmark for Multi-step Reasoning
(DABstep) which comprises over 450 authentic data analysis tasks derived directly from operational
workloads at Adyen. Distinctly, these tasks combine structured (e.g., CSV, JSON) and unstruc-
tured data (e.g., text, domain-specific documentation or complicated manuals), requiring agents to
demonstrate technical data manipulation skills (spanning SQL, statistical analysis, coding), a deep
understanding of contextual instructions, and the ability to plan hierarchically.

A core design principle of DABstep is its focus on multi-step reasoning complexity. Unlike bench-
marks where tasks might be solvable via single-shot generation [47, 24, 21, 20], *hard tasks in DAB-
step are designed to require multi-step reasoning*. Agents must decompose problems into sequential,
iterative steps—such as filtering data, computing aggregates, consulting reference tables, and han-
dling intermediate results—often requiring interaction with a code execution environment as shown

---

♣ Equal contribution.

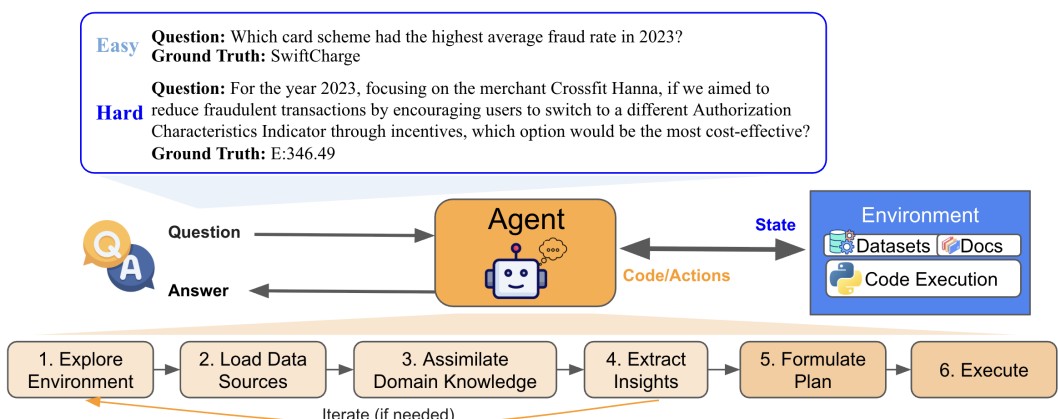

Figure 1: System overview of DABstep's agent-task interaction. The figure illustrates the core components: task input (questions), agent, execution environment, and output (answers). Two representative questions are shown: a Risk/Fraud question from the Easy set (top), requiring 2+ data sources and at least 3 execution steps; and a Scheme Fees question from the Hard set (bottom), requiring 3+ sources and more complex reasoning over at least 6 steps. Agents must combine contextual understanding, code execution, and iterative refinement to produce a correct answer. See full trace example in section A.4

in Figure 1. Furthermore, DABstep is designed for accessibility with a low barrier to entry, avoiding complex scaffolding or environments. An automated online leaderboard facilitates easy submission and *standardized* evaluation while also fostering community participation.

Our evaluations underscore DABstep's significant challenge to current state-of-the-art LLMs. Top-performing agents, such as o4-mini [34], at time of writing, achieve only 14.55% accuracy (Table 1), highlighting substantial gaps between current agent capabilities and the demands of rigorous, practical data analysis.

This paper makes the following key contributions:

- **Data**: A novel benchmark featuring **over 450 real-world data analysis tasks** designed to test complex, multi-step reasoning and planning while leveraging diverse data sources including a large (+100k) payments dataset among others.

- **Factoid Evaluation Framework**: An objective and standardized evaluation methodology centered on factoid answers with binary (right/wrong) outcomes, supported by a flexible scoring mechanism to handle formatting variations *fairly*.

- **Baselines**: Performance results and failure modes for leading open and closed LLM agents, identifying current limitations and key areas for future research.

- **Community Platform**: An accessible setup including a developer set, a quick-start notebook, an open-source baseline code, and a centralized live leaderboard to track progress and foster collaboration.

Through DABstep, we aim to drive progress in developing AI agents capable of rigorous, practical, multi-step data analysis, better aligned with real-world analytical needs.

## 2 Design Principles & Related Works

The guiding philosophy behind DABstep is grounded in four key principles that collectively emphasize realism, complexity, objectivity, and accessibility. In Table 2 we provide a high-level comparison to other related benchmarks on the main characteristics that we consider required to evaluate real-world data analysis tasks.

Table 1: Performance of baseline models on the DABstep benchmark (Hidden Test Set). Scores reflect accuracy (%) on Hard and Easy splits. Costs are estimates based on public pricing at the time of evaluation (Q1 2025) and token usage; open models run locally are considered free ('-'). All baselines run for a maximum of 10 steps per task with a ReAct style prompt except for the reasoning models. See Section 4 for methodology.

| Name | Hard (%) | Easy (%) | Total Cost ($) |
|---|---|---|---|
| o4-mini [34] | 14.55 | 76.39 | 93 |
| Claude 3.7 Sonnet [4] | 13.76 | 75.00 | 139 |
| o3-mini [34] | 13.76 | 72.22 | 85 |
| Gemini 2.5 Pro [10] | 12.70 | 66.67 | 270 |
| GPT 4.1 [33] | 12.43 | 80.56 | 155 |
| o1 [32] | 11.11 | 69.44 | 435 |
| Deepseek R1 [12] | 11.04 | 68.21 | 3 |
| Claude 3.5 Sonnet [3] | 9.26 | 77.78 | 97 |
| Llama 4 Maverick [2] | 8.73 | 75.00 | - |
| GPT-4o [30] | 6.08 | 66.67 | 53 |
| Deepseek V3 [26] | 5.56 | 66.67 | 2 |
| Claude 3.5 Haiku [3] | 5.03 | 77.78 | 35 |
| Llama 3.3 70B [11] | 3.70 | 68.06 | - |
| GPT-4o-mini [31] | 3.44 | 69.44 | 3 |
| Llama 4 Scout [2] | 1.85 | 52.78 | - |
| Llama 3.2 1B [1] | 0.00 | 1.39 | - |

## 2.1 Real-World and Multi-Step Complexity

A central design principle of DABstep is its emphasis on realistic analytical challenges that require multi-step reasoning over heterogeneous data sources. Unlike benchmarks that focus on abstract math problems [14], isolated code snippets [7], or synthetic QA tasks [8], DABstep's over 450 tasks are derived directly from operational workloads at Adyen, reflecting the complex, iterative problem-solving scenarios faced by professional data analysts.

These tasks are grounded in real-world financial analysis and integrate both structured data (e.g., CSV tables like `payments.csv`, JSON files like `fees.json`) and unstructured documentation (e.g., Markdown files like `manual.md`). Solving them demands technical proficiency in data manipulation (e.g., filtering, aggregation, joins), statistical reasoning, and the ability to extract and apply domain-specific rules from documentation. For example, agents must answer questions such as "Which card scheme had the highest average fraud rate in 2023?" or perform scenario analysis like "If merchant X changed its business category, how would that affect fees?"

Crucially, DABstep tasks are explicitly designed to resist one-shot code generation strategies [47, 24]. Especially in the Hard split (84% of tasks), no question can be answered through a single direct execution. Hard tasks require iterative data filtering and cross-referencing, which single-shot code cannot handle. Instead, agents must follow a multi-step reasoning process that involves identifying relevant context, synthesizing data across files, computing intermediate results, and validating outputs. Many tasks require cross-referencing structured sources such as tables (CSV or otherwise) or JSON datasets with unstructured content such as documentation or technical manuals, and executing multi-stage plans within a Python environment (see Figure 1). Previous code generation benchmarks [21, 7, 25] focus on isolated coding skills but lack the requirement for multi-step reasoning grounded on domain knowledge across structured and unstructured sources. Other benchmarks moved towards iterative analysis via interactive environments with multiple code executions inspired by Intercode [44], but lacked the focus on domain-specific knowledge integration during planning [23, 17], are limited to text-to-Pandas within Jupyter environments [46] or do not require integrating with heterogeneous data sources [15].

This emphasis on sequential, tool-augmented reasoning distinguishes DABstep from benchmarks focused solely on Text-to-SQL benchmarks [47, 24, 40, 13, 22, 50] or closed-domain QA [18], and better reflects the iterative nature of real-world data workflows. As evidenced in Section 4, even state-of-the-art LLM agents struggle with these challenges, especially when planning, tool use, or implicit instruction following is required.

Table 2: Comparison with existing related benchmarks. Columns include the benchmark topic (Topic), the number of tasks (# Tasks) and whether the tasks in the benchmark involve: integrating heterogeneous data sources (Hetero.), come from real-world scenarios driving business value and not from just educational, synthetic or online resources (Real World), require following domain-specific business knowledge and/or rules to arrive to a solution (Domain Knowledge), require multi-model input handling (Multi-modal), require multiple steps of reasoning to arrive to a solution (Multi-step), require agents to perform analysis via code (Code), can be objectively evaluated (Objective Evals). DABstep v1 focuses on structured/unstructured text only.

| Benchmark | Topic | Hetero. | Real World | Domain Knowledge | Multi-modal | Multi-step | Code | Objective Evals | #Tasks |
|---|---|---|---|---|---|---|---|---|---|
| FinanceBench [18] | Finance QA | ✓ | ✓ | ✓ | ✗ | ✗ | ✗ | ✗ | 10,231 |
| GAIA [28] | General QA | ✓ | ✓ | ✗ | ✓ | ✗ | ✓ | ✓ | 466 |
| MATH [14] | Math QA | ✗ | ✗ | ✗ | ✗ | ✗ | ✗ | ✓ | 12,500 |
| GSM8K [29] | Math QA | ✗ | ✗ | ✗ | ✗ | ✗ | ✗ | ✓ | 8,500 |
| Spider [47] | Text-to-SQL | ✗ | ✗ | ✗ | ✗ | ✗ | ✗ | ✓ | 1,181 |
| Spider 2 [23] | Text-to-SQL | ✓ | ✗ | ✓ | ✗ | ✓ | ✓ | ✓ | 632 |
| BIRD [24] | Text-to-SQL | ✗ | ✗ | ✓ | ✗ | ✗ | ✗ | ✓ | 12,751 |
| KaggleDBQA [22] | Text-to-SQL | ✓ | ✗ | ✗ | ✗ | ✗ | ✓ | ✓ | 272 |
| WikiSQL [50] | Text-to-SQL | ✗ | ✓ | ✗ | ✗ | ✗ | ✗ | ✓ | 80,654 |
| HumanEval [7] | Text-to-Python | ✗ | ✗ | ✗ | ✗ | ✗ | ✓ | ✗ | 164 |
| NL2Bash [25] | Text-to-Bash | ✗ | ✓ | ✗ | ✗ | ✗ | ✗ | ✗ | 9,305 |
| Arcade [46] | Text-to-Pandas | ✗ | ✗ | ✗ | ✗ | ✓ | ✓ | ✓ | 10,082 |
| SWE-Bench [19] | Software | ✗ | ✓ | ✗ | ✓ | ✗ | ✗ | ✓ | 2,294 |
| WebArena [51] | Web | ✓ | ✗ | ✗ | ✓ | ✓ | ✗ | ✓ | 812 |
| OSWorld [42] | Computer Control | ✓ | ✓ | ✗ | ✓ | ✓ | ✗ | ✓ | 369 |
| Intercode [44] | Iterative code | ✗ | ✗ | ✗ | ✗ | ✓ | ✓ | ✓ | 1351 |
| MLAgentBench [16] | Machine Learning | ✓ | ✓ | ✗ | ✓ | ✓ | ✓ | ✓ | 13 |
| DABench [15] | Data Analysis | ✗ | ✓ | ✗ | ✗ | ✓ | ✓ | ✓ | 257 |
| DA-Code [17] | Data Science | ✓ | ✓ | ✗ | ✗ | ✓ | ✓ | ✓ | 500 |
| DS-1000 [21] | Data Science | ✗ | ✓ | ✗ | ✗ | ✗ | ✓ | ✓ | 1,000 |
| Spider2-V [5] | Data Science | ✓ | ✓ | ✓ | ✗ | ✗ | ✓ | ✓ | 494 |
| DSEval [48] | Data Science | ✗ | ✗ | ✗ | ✗ | ✓ | ✓ | ✓ | 825 |
| DSBench [20] | Data Science | ✓ | ✗ | ✗ | ✓ | ✓ | ✓ | ✗ | 540 |
| DABstep (ours) | Data Analysis | ✓ | ✓ | ✓ | ✗ | ✓ | ✓ | ✓ | 450 |

## 2.2 Objective Evaluation

DABstep was designed with a guiding principle of verifiable answers to ensure robust and straightforward evaluations. Unlike many benchmarks that include tasks validated by LLM-as-a-judge approaches [20, 43, 28], we selectively curated questions to produce objective, factoid answers—such as numbers, lists, or concise strings—amenable to automated scoring. This choice reflects a deliberate trade-off: while many impactful use-cases often involve free-form outputs (e.g., analytical reports, narrative summaries), their evaluation often requires subjective, resource-intensive methods like LLM-as-a-judge, introducing potential bias and infrastructure costs [49]. By focusing on verifiable answers, DABstep sacrifices some task diversity (like more open-ended tasks with free-form output) but gains in advantages: high evaluation reliability, scalability without LLM dependency, and a streamlined experience for developers and users. As a concrete example, factoid task answers may be numerical (e.g., 42) or a list of items (e.g., 'CardA, CardB'). This curation aligns with our goal of creating a benchmark that prioritizes reliability and accessibility, enabling fair comparisons of LLM agents on data analysis tasks without the overhead of complex evaluation pipelines. However, even factoid answers exhibit variability (e.g., '42' vs. 'forty-two' vs. '42.00'), necessitating a flexible and deterministic scoring mechanism (detailed in Section A.2) to maintain fairness without reverting to rigid exact matching or LLM-based subjectivity.

## 2.3 Simple accessible setup

A key design philosophy underpinning DABstep is ensuring a low barrier to entry for researchers and practitioners. We deliberately avoided requiring complex environments or specialized infrastructure often associated with other benchmarks, like general agent platforms [27, 5, 9, 42, 41], software and

ML engineering development benchmarks [19, 6, 16] and dedicated SQL benchmarks [47]. Instead, DABstep is designed for ease of use, requiring only a standard Python runtime for task execution.

The evaluation process is streamlined through an automated online leaderboard. Participants can submit their results easily, receiving *standardized* evaluations without needing to manage complex local evaluation setups and avoids biased cherry-picked baselining [35]. To further support accessibility, baseline implementations and prompts are provided openly. This simplicity in setup and evaluation is intended to encourage broad participation from the research community, mirroring the accessibility that has spurred progress in widely adopted NLP benchmarks [39]. By minimizing setup complexity, DABstep allows researchers to focus directly on the core challenges of multi-step reasoning and data analysis for LLM agents.

## 3 Benchmark

### 3.1 Task Curation

DABstep consists of over 450 tasks derived from real-world analytical challenges at Adyen, curated to evaluate agent capabilities through factoid-style question answering with objective scoring. Each task mirrors typical workflows and queries faced by professional data analysts. The tasks were selected from real but anonymized internal queries to ensure a diverse range of data manipulation, reasoning steps, and contextual understanding. Each task (see example in Section A.4) is presented with a natural language prompt, which includes:

- **Question**: A specific question (e.g., "Which card scheme had the highest average fraud rate in 2023?") or an analytical scenario (e.g., assessing the fee impact of a merchant changing business categories).

- **Guidance**: Clear guidance on the required answer format (e.g., "Provide the name of the scheme" or "Provide the result as `scheme:fee` where fee is rounded to 2 decimal places"). This minimizes penalties due to trivial formatting errors.

- **Context**: A set of context files (datasets and documentation) necessary to solve the task.

- **Level**: A difficulty tag (Easy or Hard).

The tasks require agents to integrate information from heterogeneous data sources, demanding both technical data analysis skills (using code) and domain-specific understanding learned from the provided context.

To provide the necessary context or domain knowledge for agents to reason over while solving tasks, we release several datasets, including a large payments dataset with over 100,000 anonymized transactions and various industry-specific manuals and documentation represented in simplified formats. For instance, in the finance industry, business context is often outlined in extensive handbooks from payment networks, regulators, and processors. For this benchmark version, we have created markdown documentation distilling essential business knowledge (e.g., concepts like Merchant Category Codes (MCC), Authorization Characteristics Indicators (ACI), scheme fee structures) into a precise yet accessible format crucial for solving many tasks accurately.

**Symbolic Reasoning through Parameterization** In the spirit of GSM-Symbolic [29], many base task types have been expanded into multiple concrete instances by systematically varying parameters like time ranges, merchant names, or specific thresholds. This parameterization significantly increases the number of unique task instances derived from a smaller set of core analytical workflows. *Concretely, out of the 450 total questions, 95 of those are core questions.* The rationale is twofold: first, to minimize the possibility of agents succeeding through 'lucky guesses' or memorization of answers potentially seen during pre-training; second, to rigorously validate the core reasoning capabilities of the agents – their ability to apply the same logical steps consistently across different input values, thereby testing generalization. This approach emphasizes evaluating the underlying problem-solving process rather than just retrieving specific facts. A concrete example: A task asking for fraud rates in September is varied by changing the month to July or February, or by altering merchant names.

## 3.2 Task Distribution

DABstep comprises over 450 data analysis tasks, permuted from a core of 95 (see 3.1), combining structured datasets (like CSV tables and JSON files) with unstructured text (Markdown documentation). The tasks are categorized by difficulty:

- **Easy Tasks:** 72 tasks (approximately 16% of the total). These generally involve querying or processing a single primary dataset with minimal reliance on complex contextual information from documentation. They serve as basic sanity checks or warm-ups.

- **Hard Tasks:** 378 tasks (approximately 84% of the total and permuted from 23 core questions). These tasks form the core challenge of the benchmark. They typically require cross-referencing multiple data sources, understanding domain-specific concepts explained in manuals/documentation and executing a multi-step reasoning process involving several stages of data manipulation and interpretation. Section A.4 shows that solving these tasks goes significantly beyond simple single-shot code generation capabilities.

This distribution, heavily weighted towards Hard tasks, intentionally reflects the complexity of real-world data analysis challenges faced by professional data analysts, demanding robust technical skills combined with planning and reasoning capabilities. From our baselines in Section 4, there is a 49% correlation with performance on the easy set to performance on the hard set. In Section A.3 we provide a snapshot of the key data sources included in the benchmark's context, illustrating the mix of structured formats and unstructured documentation.

## 3.3 Evaluation Protocol

DABstep is designed for automated, fast, and factual evaluation. Echoing the principles of benchmarks like GAIA [28], each task requires a specific factoid answer: typically a string (one or a few words), a number, or a list of strings/numbers (comma-separated unless otherwise specified in the guidance). There is only one correct ground truth answer for each task.

Evaluation is performed via automated comparison between the agent's final answer and the corresponding ground truth answer, using a flexible scoring algorithm detailed in Section A.2 (and Algorithm 1 in the Appendix). This quasi-exact match approach, with type-specific normalization and tolerance, ensures objective and scalable scoring.

**Hidden Test Set for Zero-Shot Generalization**   To robustly evaluate the zero-shot generalization capabilities of agents, DABstep employs a single, held-out hidden test set. This set contains the majority of the benchmark tasks and is used exclusively for the official evaluation conducted via our public leaderboard. We do not release separate public validation or test sets derived from this hidden data.

This design choice serves several purposes. First, it encourages the development of agents that generalize well across the diverse range of data analysis tasks represented in the benchmark, rather than overfitting to a specific public subset. Second, by hiding the test ground truths, we maintain the long-term integrity of the benchmark and reduce the risk of leakage into the training corpora of future models, i.e *saturation*. This safeguard ensures that DABstep remains a reliable measure of true generalization capabilities over time. Finally, our design aligns with best practices for rigorous benchmarking [35], supporting fair and standardized evaluation across all participants.

To facilitate development, local testing, and environment setup without requiring interaction with the official leaderboard for every iteration, we release a smaller public developer set. This set includes a representative sample of tasks with their ground truth answers, allowing researchers to verify their agent implementations and scoring logic end-to-end before submitting to the leaderboard for evaluation on the hidden test set.

## 4 Baselines

To establish initial performance levels on DABstep and highlight the gap between current LLM agent capabilities and the demands of complex, real-world data analysis tasks, we evaluated a range

of state-of-the-art open and closed-source language models available at the time of evaluation (Q1 2025).

## 4.1 Baseline Setup

To ensure fair and scalable evaluation across a diverse range of models, we adopted a standardized setup with minimal model-specific tuning. Most models were prompted using a generic ReAct-style approach [45], framing the LLM as a data analyst that reasons step by step, invokes a Python execution tool when needed, and formats its final answer in a structured output. The prompt included abstract demonstrations of the desired reasoning-action-observation loop. We intentionally avoided heavily engineered or model-specific prompts to keep comparisons consistent across different architectures.

For a few models—such as o4-mini, o3-mini, o1, R1 and Gemini 2.5 Pro—we employed a "Reasoning Prompt," a slightly adapted variant better aligned with their internal reasoning paradigms. However, these were still standardized across those models to ensure internal consistency.

Open-source models were run on a dedicated cluster with 4× Nvidia A100 GPUs (80GB each). All tasks were executed in isolated environments where the agent had access to a Python kernel, enabling dynamic code execution for data loading, manipulation, and statistical analysis via standard libraries. Each task environment also included mounted context files containing relevant datasets and documentation.

Notably, we avoided the use of complex agent frameworks or external orchestration layers. The entire setup consisted of a lightweight wrapper [36] around the LLM API, providing Python execution and I/O, but leaving all decision-making—including reasoning, code generation, and result interpretation—to the model itself. This minimal infrastructure ensures that observed performance reflects the model's inherent capabilities, rather than the sophistication of external tooling.

The full baseline implementation, including prompts and evaluation code, is available in the benchmark repository.*

## 4.2 Results

Table 1 presents accuracy results, broken down by the Easy (16% of tasks) and Hard (84% of tasks) splits of the benchmark. These scores demonstrate the significant challenge posed by DABstep, particularly in the Hard split. Even the top-performing model, o4-mini (using the reasoning prompt) achieves only 14.55% accuracy on the Hard tasks. Several other leading models—including proprietary models (e.g., Claude 3.7 Sonnet and o3-mini), as well as strong open models— scored below 14% on the Hard set.

In contrast, performance on the Easy split is considerably higher. For instance, o4-mini reaches 76.39% accuracy, suggesting that many LLMs can already handle one-shot analysis tasks with high effectiveness. The stark drop in scores on the Hard split reveals that tasks requiring multiple steps of reasoning remain largely unsolved. Notably, smaller or less specialized models, such as Llama 3.2 1B, struggle across the board, particularly on the Hard tasks, further highlighting the gap in current capabilities.

The evaluation also considered the approximate cost of running the benchmark for proprietary API-based models (Table 1), revealing significant trade-offs between performance and cost. Costs per full benchmark run varied widely, highlighting the economic factors involved in deploying these agents.

## 4.3 Failure Modes

Agent performance often degrades when deviating from the ideal iterative trajectory outlined in Figure 1, particularly during planning and execution phases. Analysis of incorrect agent trajectories revealed common failure patterns frequently linked to DABstep's design principles emphasizing multi-step complexity and real-world nuances. Section A.4 illustrates a real agent trace example with some of the failure modes we introduce below.

---

*https://huggingface.co/spaces/adyen/DABstep/tree/main/baseline

**Planning and Instruction Following Deficiencies.** Agents often struggled to correctly decompose complex Hard tasks into a viable sequence of sub-steps. They might miss necessary intermediate calculations, fail to consult required documentation at the right point, or hallucinate incorrect analysis plans. A common issue was attempting calculations before reading the relevant sections of documentation that defined key domain terms or logic. In addition, we observed that agents tend to perform well at following instructions which are **explicitly** stated in the context (i.e domain-specific formula). However, agents are considerably more prone to fail when they face a task in which they need to follow rules **implicitly** mentioned in the context (i.e domain-specific rule with multiple downstream implications) or composite rules which are linked together implicitly. While our observations are preliminary, one plausible explanation for these failure modes lies in the nature of the current retrieval systems that can be scaffolded around models and the inductive biases of LLMs. Specifically, the self-attention mechanism [38], central to the LLM architecture and retrieval systems, primarily captures semantic similarity (i.e., relationships based on token co-occurrence and contextual proximity) rather than abstract conceptual similarity (i.e., relationships between underlying ideas irrespective of their surface-level expression). This might result in agents missing to link together pieces of information, not explicitly mentioned, which are crucial for the task at hand. In contrast, human analysts excel at identifying and linking such implicit cues, highlighting a potential area where current models fall short.

**Inefficient Code.** We observe that the code generated by the agent becomes increasingly inefficient as the complexity of the reasoning required by a task also increases. In particular, agents tend to default to low-level constructs such as explicit for-loops for tasks like computing group averages, even when high-level abstractions or idioms such as group-by or filter are available. This suggests that while the models can produce functionally correct code, their reasoning process fails to generalize to more abstract or idiomatic programming patterns as task difficulty grows.

**Multi-step Instruction Following.** In addition to the main, user-provided agent prompt, answers to tasks must also follow a formatting guidance prompt (Section 3.1). Potentially resulting in a complex, but realistic set of instructions to follow. Studies have shown [37] that enforcing multiple instructions in a single prompt, such as, in this case, formatting guidelines, significantly reduces LLM reasoning abilities. We did observe a non-trivial amount of errors resulted from failing to follow the specified output format guidance, for example, agents will often provide conversational text instead of outputting a number or apply incorrect rounding, wrong list delimiters or order, suggesting decomposing instructions into independent units as in [37].

**Prompt Sensitivity.** Certain models known for strong reasoning capabilities (R1, o1, etc) performed poorly with the standardized ReAct prompt (some scoring near 0% initially before trying a reasoning-specific standard prompt), indicating a high sensitivity to prompt structure and a potential reliance on specific custom prompting techniques not used in our fair, standardized baseline evaluation.

While these baseline results represent performance with non-optimized, standardized prompts and **should be considered a lower bound**, they effectively demonstrate the significant challenges DAB-step presents in areas crucial for practical data analysis: robust multi-step reasoning, accurate handling of diverse data sources and domain knowledge, reliable tool use, and precise instruction following. These findings establish a clear benchmark and highlight key areas for future research aimed at improving agent capabilities.

## 5 Conclusion, Limitations & Future Work

DABstep is a large-scale benchmark designed to rigorously evaluate autonomous agents on realistic, multi-step data analysis tasks. With over 450 grounded challenges derived from financial workloads, DABstep uniquely combines code execution, contextual reasoning over structured and unstructured data, and an objective factoid-based evaluation protocol. Our baseline results show a stark capability gap: state-of-the-art LLM agents achieve only $14.55\%$ accuracy on Hard tasks, underscoring critical limitations in reasoning, planning, tool use, and instruction following. By releasing data, evaluation tools, baseline code, and a public leaderboard, we aim to foster community engagement and accelerate research on practical, agentic data analysis.

However, this is the first iteration of the DABstep benchmark. It currently supports only text-based inputs and factoid-style outputs, limiting its ability to assess critical skills like visual reasoning (e.g., from charts or PDFs) or open-ended analysis (e.g., narratives, recommendations). These constraints reflect deliberate trade-offs to favor objective, scalable evaluation, but future iterations will address them by (i) expanding financial tasks (e.g., approval rate and temporal analysis), (ii) incorporating other domains (e.g., healthcare, e-commerce), (iii) increasing data scale and heterogeneity (e.g., longer documents, PDF manuals), (iv) introducing multimodal tasks, and (v) pushing toward agents capable of interactive clarification, exploratory analysis, and synthesis. These directions aim to close the gap between benchmark and real-world analytical workflows, while continuing to emphasize transparency, rigor, and accessibility.

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

# A  Technical Appendices and Supplementary Material

## A.1  Usage and Accessibility

DABstep is designed for broad accessibility. The benchmark platform, including baseline code, documentation, and the live leaderboard, is publicly hosted on **Hugging Face Spaces**. The core dataset, anonymized and released under an open license [Creative Commons Attribution 4.0 International], is available on the **Hugging Face Hub** with a detailed dataset card. The platform provides a standardized environment for evaluation via leaderboard submissions. [†]

---

[†]https://creativecommons.org/licenses/by/4.0/

## A.2 Hybrid Scoring Algorithm

To ensure objective and robust evaluation while accommodating minor, semantically irrelevant variations in agent outputs, DABstep employs a hybrid scoring algorithm. This approach avoids the brittleness of pure exact string matching, which would unfairly penalize correct answers with trivial differences (e.g., "\$42.00" vs. "42"), and sidesteps the potential biases, costs, and complexities associated with LLM-as-a-judge methods [49].

The algorithm first normalizes both the agent's predicted answer and the ground truth answer (e.g., converting to lowercase, trimming whitespace). It then applies type-specific comparison logic:

- **Numeric Comparison**: If both inputs can be interpreted as numbers, it extracts the numeric values, ignoring formatting like currency symbols or thousands separators (e.g., '\$1,234.56' becomes 1234.56). Comparison allows for a small tolerance (e.g., $10^{-4}$) to handle potential floating-point rounding differences.

- **List Comparison**: If inputs appear to be lists (based on delimiters like commas or semicolons), they are split into elements. Each element is normalized (e.g., whitespace trimmed, converted to lowercase). The lists are then compared, typically ignoring the order of elements unless the task specifically requires ordered output. Normalization handles variations like 'uber, spotify, nike' matching 'Nike, Uber, Spotify'. Recursive calls to the scoring function handle comparisons of elements within the lists if they are complex (e.g., lists of numbers).

- **String Comparison**: For general string answers, basic cleaning is applied (e.g., removing punctuation, extra whitespace). If the cleaned strings match exactly, the answer is correct. If not, fuzzy string matching (e.g., using Levenshtein distance or a similar metric) is employed, accepting answers with a high similarity score (e.g., $> 0.95$) to the ground truth which handles minor typos or variations. Special handling might apply for single-word vs. multi-word comparisons.

This tiered approach ensures that the evaluation focuses on the semantic correctness of the factoid answer, providing a fair yet rigorous assessment aligned with DABstep's goal of objective, automated scoring. Pseudocode for the algorithm is provided in Algorithm 1 in the Appendix (Figure A.2).

To validate our automated scoring algorithm, we collected 75 model-generated answers across Easy and Hard tasks from a diverse set of LLMs (GPT-4, Claude 3.5, LLaMA 3). Each answer was manually judged by two independent annotators, who labeled the response as correct or incorrect according to the task instructions and gold reference. Inter-annotator agreement was 97.3% (Cohen's k = 0.94), with disagreements resolved via discussion.

Our scoring function matched human judgment on all 75 examples, achieving 100% accuracy. Using binomial confidence intervals, this yields a 95% CI of [96.2%, 100%], indicating strong reliability. Crucially, many examples required tolerance to numeric rounding, flexible list ordering, or fuzzy string similarity—highlighting the importance of a robust scoring mechanism.

## A.3 Dataset Snapshot

In Table 3 we provide a concise overview of each file in the DABstep benchmark's dataset bundle. This snapshot highlights file names, formats, and a brief description of their contents, which together capture the key aspects of our finance use cases.

**Algorithm 1** Hybrid Answer Scoring Algorithm (Pseudocode)

---

1: **procedure** SCOREANSWER(predicted_answer, ground_truth)
2:     $pred \leftarrow$ Normalize(`predicted_answer`)            ▷ e.g., lowercase, trim whitespace
3:     $gt \leftarrow$ Normalize(`ground_truth`)
4:     **if** IsNumeric($pred$) **and** IsNumeric($gt$) **then**         ▷ Check if both are numeric
5:         $n_{pred} \leftarrow$ ExtractNumeric($pred$)              ▷ Handles , etc.
6:         $n_{gt} \leftarrow$ ExtractNumeric($gt$)
7:         **return** CompareNumeric($n_{pred}$, $n_{gt}$, tolerance=$10^{-2}$)     ▷ Allows small float diff
8:     **end if**
9:     **if** IsList($pred$) **and** IsList($gt$) **then**         ▷ Check if both look like lists
10:         $l_{pred} \leftarrow$ SplitSortNormalizeList($pred$)     ▷ Split, normalize elements, sort
11:         $l_{gt} \leftarrow$ SplitSortNormalizeList($gt$)
12:         **if** length($l_{pred}$) $\neq$ length($l_{gt}$) **then**
13:             **return** `false`
14:         **end if**
15:         all_match $\leftarrow$ `true`
16:         **for** i from 0 to length($l_{pred}$) - 1 **do**
17:             **if not** ScoreAnswer($l_{pred}[i]$, $l_{gt}[i]$) **then**     ▷ Recursive call for elements
18:                 all_match $\leftarrow$ `false`
19:                 **break**
20:             **end if**
21:         **end for**
22:         **return** all_match
23:     **end if**

                        ▷ Default to string comparison
24:     $pred_{clean} \leftarrow$ CleanString($pred$)     ▷ Remove punctuation, extra spaces
25:     $gt_{clean} \leftarrow$ CleanString($gt$)
26:     **if** $pred_{clean} = gt_{clean}$ **then**
27:         **return** `true`             ▷ Exact match after cleaning
28:     **end if**
29:     similarity_score $\leftarrow$ CalculateStringSimilarity($pred_{clean}$, $gt_{clean}$)   ▷ e.g., Levenshtein ratio
30:     **if** similarity_score $> 0.95$ **then**         ▷ Threshold for fuzzy match
31:         **return** `true`
32:     **end if**
33:     **return** `false`             ▷ Default to false if no match
34: **end procedure**
        ▷ *Helper functions like Normalize, IsNumeric, ExtractNumeric, CompareNumeric, IsList, SplitSortNormalizeList, CleanString, CalculateStringSimilarity are assumed.*

---

Table 3: Snapshot of key datasets provided as context within the DABstep benchmark, covering aspects of the financial payments sector.

| Name | Description |
| --- | --- |
| `payments.csv` | Anonymized payments dataset containing over 138,000 transactions with features relevant to fraud detection and risk analysis use-cases. |
| `payments-readme.md` | Human-readable documentation explaining the columns and content of the `payments.csv` dataset. |
| `acquirer_countries.csv` | Table mapping acquiring bank identifiers to their respective countries. |
| `fees.json` | Dataset detailing various scheme fee structures (over 1000 entries), often dependent on transaction and merchant attributes. |
| `merchant_category_codes.csv` | Table listing Merchant Category Codes (MCCs) and their descriptions. |
| `merchant_data.json` | Table containing descriptive information about various merchants (anonymized). |
| `manual.md` | A comprehensive guide (distilled for the benchmark) explaining core payment processing concepts (e.g., Account Types, MCC, ACI), detailing fee calculation logic based on merchant and transaction attributes, and outlining best practices for minimizing costs and fraud risk. Essential for solving many Hard tasks. |

## A.4 Failure Mode Example

Figures 2–11 illustrate a trace from a real task execution by a Claude 3.7 Sonnet agent, one of the best performing baselines, on the DABstep benchmark. For clarity, we omit action outputs that are excessively verbose (e.g., full documentation dumps, programming error traces).

The task requires identifying all applicable fees for a specific merchant on a given date, with the expected output being a comma-separated list as defined in the task guidelines. To address this, the agent first explores the data sources available in the `data/context` environment. In steps 0 and 1 (Figures 2 and 3), the agent succeeds in identifying and retrieving the relevant context files.

In steps 2 and 3, the agent parses these data to extract merchant business attributes (Figures 4 and 5). Next, it returns to the context in step 4 (Figure 6) to refine its understanding of fee scheme rules. After refining its understanding, at step 5 (Figure 7), the agent recognizes a missing piece of the puzzle: merchant payment traffic characteristics, which it attempts to find next.

Step 6 (Figures 8–10) represents the synthesis stage, where the agent combines insights from merchant's business and payment traffic with domain-specific rules to discriminate the applicable fees. Finally, in step 7 (Figure 11), it composes an answer conforming to the task output format.

Despite the apparently coherent strategy, the agent ultimately fails to produce the correct result. This is because the agent misses a critical piece of information from the context (in step 6, Figure 9), although it has ingested into its memory in step 1 (Figure 3). The overlooked information is that merchant monthly fraud rates and other monthly payment traffic statistics discriminate between fees, and these statistics need to be computed to then be filtered on. Therefore the provided list of fees in the answer is a superset of the actual applicable fees. This behavior is an example of *Planning and Instruction Following Deficiencies* described in Section 4.3. Step 6 (Figure 9) also shows some inefficient code where the agent iterates transactions and fees using nested for-loops instead of using libraries built-in operations like group-by and filters.

LLM: Claude 3.7 Sonnet - Input Tokens: 81k - Output Tokens: 8k

**Question**: For the 10th day of the year 2023, what are the Fee IDs applicable to Belles_cookbook_store?

**Guidelines**: Answer must be a list of values in comma separated list, eg: A, B, C. If the answer is an empty list, reply with an empty string. If a question does not have a relevant or applicable answer for the task, please respond with 'Not Applicable'
Before answering the question, reference any documentation in the data dir and leverage its information in your reasoning / planning.

**Step 0**

**Agent**

I'll solve this task by exploring the data in the `data/context` directory, understanding the structure, and then finding the Fee IDs applicable to Belles_cookbook_store for the 10th of the year 2023.

Thought: First, I need to explore the `data/context` directory to see what files are available and understand the data structure and documentation.

Code:
```py
import os

# List files in the data/context directory
files = os.listdir('data/context')
print("Files in data/context directory:")
for file in files:
    print(file)
```

**Execution Logs**

```
Files in data/context directory:
fees.json
merchant_data.json
merchant_category_codes.csv
payments.csv
payments-readme.md
acquirer_countries.csv
manual.md
```

Figure 2: Agent step 0 where environment exploration happens.

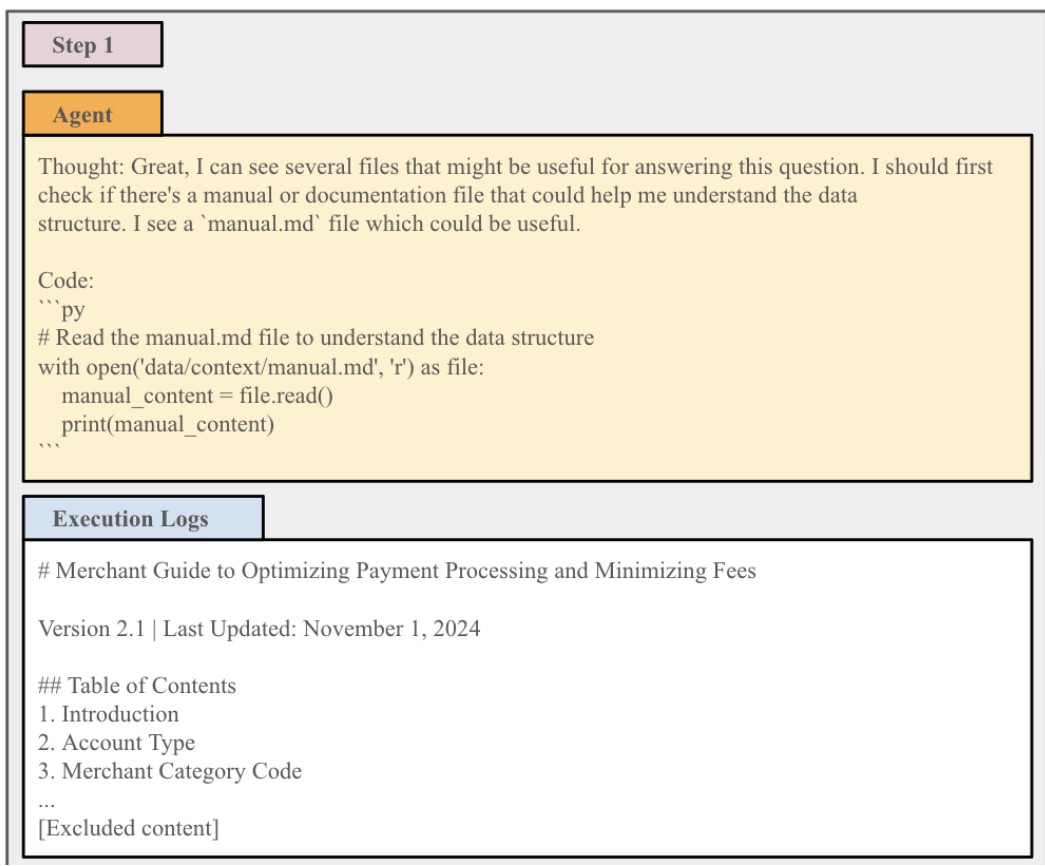

Figure 3: Agent step 1 where the agent loads domain-specific documentation into its working memory. Full documentation dump is omitted.

**Step 2**

**Agent**

Thought: The manual provides a lot of useful information. I understand that there are different files that might contain the information I need. Let's first check the `merchant_data.json` file to find information about Belles_cookbook_store.

Code:
```py
import json

# Read the merchant_data.json file
with open('data/context/merchant_data.json', 'r') as file:
    merchant_data = json.load(file)

# Check if Belles_cookbook_store is in the data
if 'Belles_cookbook_store' in merchant_data:
    print("Found Belles_cookbook_store in merchant_data.json")
    print(merchant_data['Belles_cookbook_store'])
else:
    print("Belles_cookbook_store not found in merchant_data.json")

# Let's list the keys in the merchant_data to see if the name might be formatted differently
print("\nMerchant names in the data:")
for merchant_name in merchant_data.keys():
    print(merchant_name)
```

**Execution Logs**

Belles_cookbook_store not found in merchant_data.json

Merchant names in the data:
**ERROR**: Code execution failed at line 'for merchant_name in merchant_data.keys():
    print(merchant_name)' because of the following error:
Object [{'merchant': 'Crossfit_Hanna', 'capture_delay': 'manual', 'acquirer': ['gringotts',
'the_savings_and_loan_bank', 'bank_of_springfield', 'dagoberts_vault'], 'merchant_category_code':
7997,
'account_type': 'F'}, {'merchant': 'Martinis_Fine_ ... ] has no attribute keys

Figure 4: Agent step 2 where the agent tries to find more about the merchant but it fails to produce valid code. The full error trace is omitted.

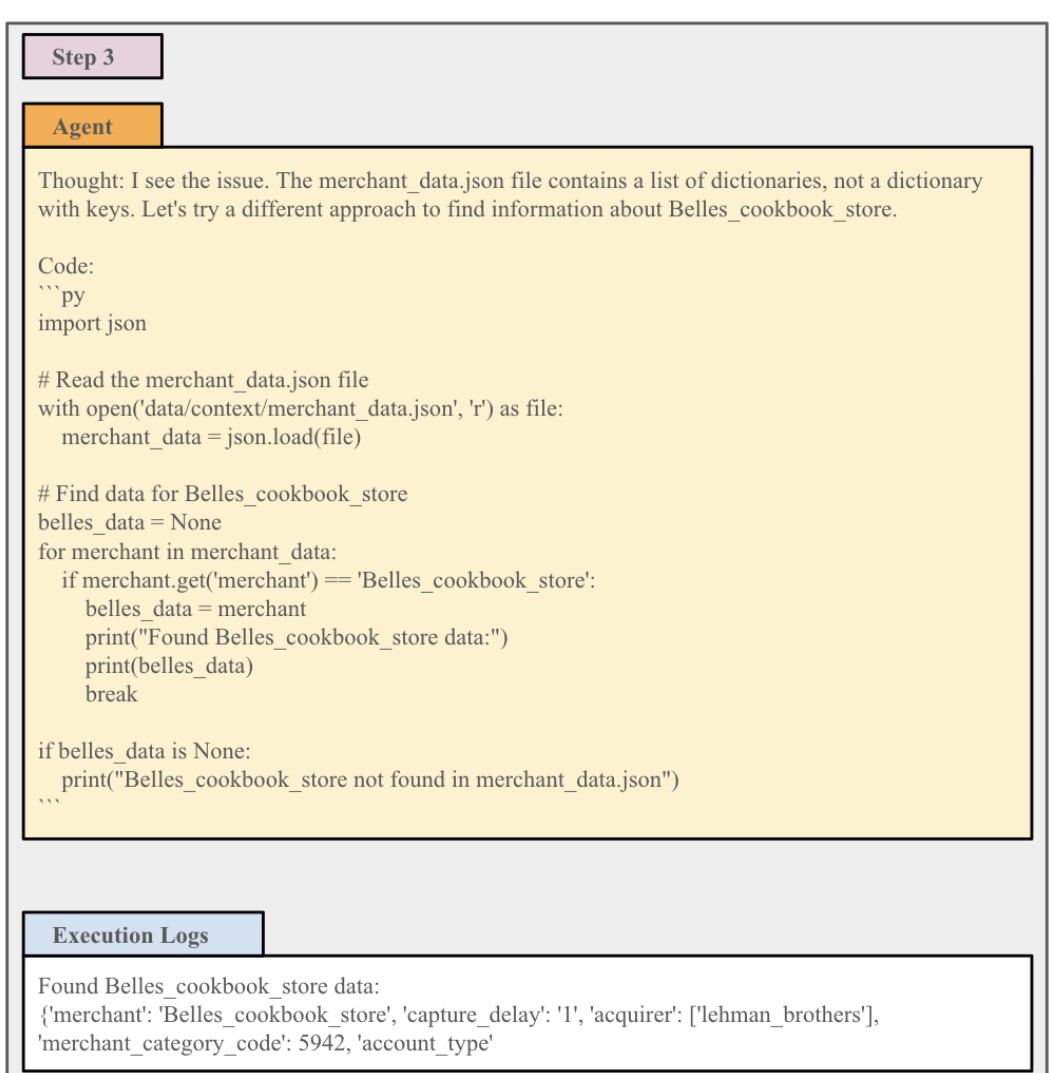

Figure 5: Agent step 3 where the agent successfully finds merchant business related data.

**Step 4**

**Agent**

Thought: Great! Now I have information about Belles_cookbook_store. According to the data, it's a merchant with category code 5942, account type 'R' (Enterprise - Retail), and a capture delay of '1'. Next, I need to look at the fees.json file to understand the fee structure and then check what fees are applicable on the 10th of the year 2023.

Code:
```py
import json

# Read the fees.json file
with open('data/context/fees.json', 'r') as file:
    fees_data = json.load(file)

# Check the structure of the fees data
print("Fees data structure:")
print(type(fees_data))
if isinstance(fees_data, list) and len(fees_data) > 0:
    print("Sample fee entry:")
    print(fees_data[0])
else:
    print("No fee data found or unexpected format")
```

**Execution Logs**

Fees data structure:
<class 'list'>
Sample fee entry:
{'ID': 1, 'card_scheme': 'TransactPlus', 'account_type': [], 'capture_delay': None, 'monthly_fraud_level': None, 'monthly_volume': None, 'merchant_category_code': [8000, 8011, 8021, 8031, 8041, 7299, 9399, 8742], 'is_credit': False, 'aci': ['C', 'B'], 'fixed_amount': 0.1, 'rate': 19, 'intracountry': None}

Figure 6: Agent step 4 where the agent refines its understanding about fee calculations.

**Step 5**

**Agent**

Thought: Now I understand the structure of the fees.json file. I need to find the fees that are applicable to Belles_cookbook_store on the 10th of the year 2023. Let me first check if there's a payments.csv file that might contain payment information for the date in question.

Code:
```py
import pandas as pd
# Read payments.csv
try:
    payments_df = pd.read_csv('data/context/payments.csv')
    print("Payments.csv columns:")
    print(payments_df.columns.tolist())

    # Let's check if there are any records for Belles_cookbook_store on the 10th of the year 2023
    if 'merchant' in payments_df.columns and 'date' in payments_df.columns:
        belles_payments = payments_df[(payments_df['merchant'] == 'Belles_cookbook_store') &
                            (payments_df['date'].str.contains('2023') &
                             payments_df['date'].str.contains('10'))]
        if not belles_payments.empty:
            print("\nPayments for Belles_cookbook_store on the 10th of 2023:")
            print(belles_payments.head())
        else:
            print("\nNo payments found for Belles_cookbook_store on the 10th of 2023")
    else:
        print("\nRequired columns 'merchant' or 'date' not found in payments.csv")
except Exception as e:
    print(f"Error reading payments.csv: {e}")

# Let's also check the payments-readme.md
try:
    with open('data/context/payments-readme.md', 'r') as file:
        readme_content = file.read()
        print("\nPayments-readme.md content:")
        print(readme_content)
except Exception as e:
    print(f"Error reading payments-readme.md: {e}")
```

**Execution Logs**

Payments.csv columns:
['psp_reference', 'merchant', .. [Excluded]]
Required columns 'merchant' or 'date' not found in payments.csv

Payments-readme.md content:
This is documentation for the payments.csv dataset [Excluded] …

Figure 7: Agent step 5 where the agent is looking for relevant information in the merchant's payment traffic.

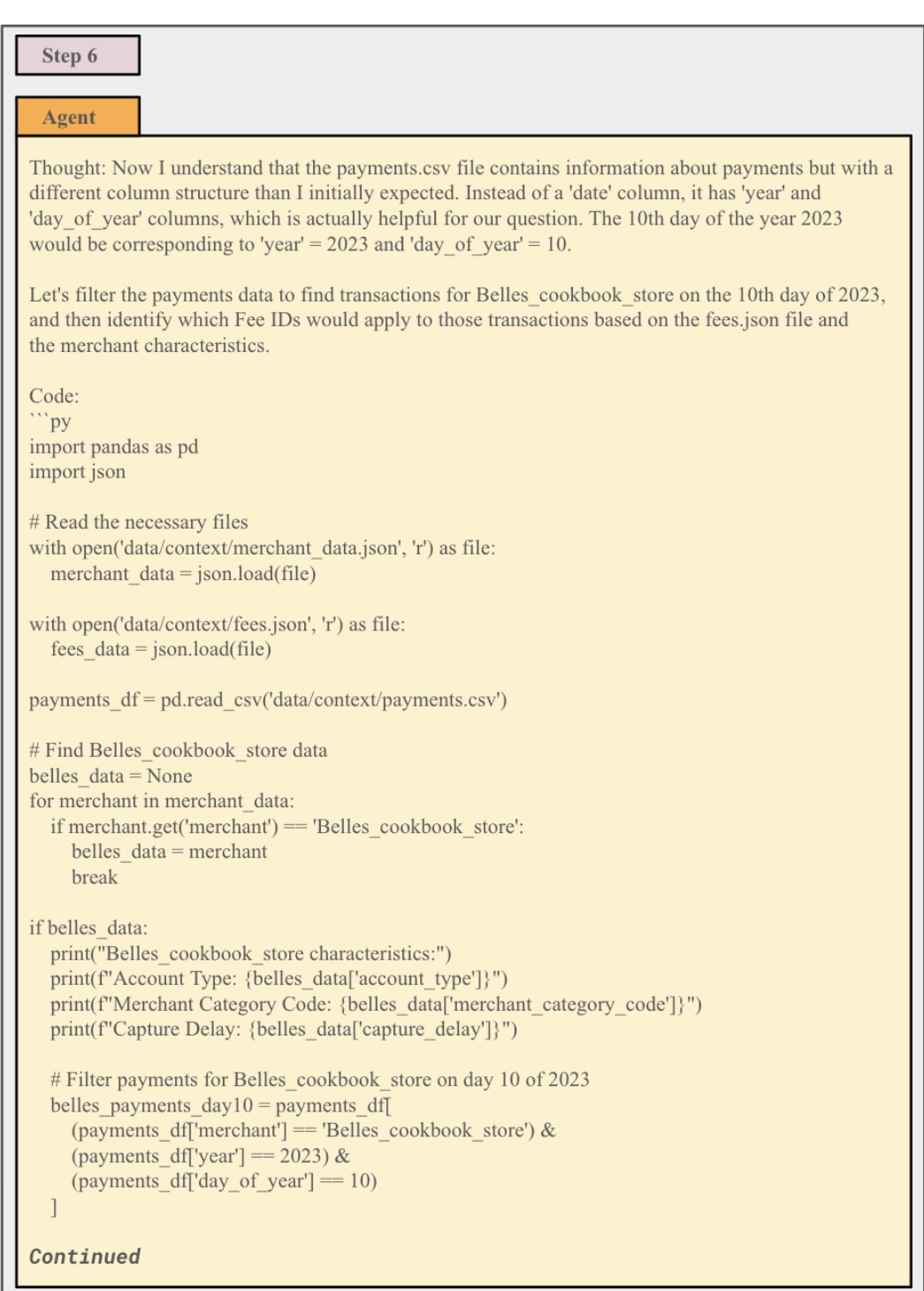

**Step 6**

**Agent**

Thought: Now I understand that the payments.csv file contains information about payments but with a different column structure than I initially expected. Instead of a 'date' column, it has 'year' and 'day_of_year' columns, which is actually helpful for our question. The 10th day of the year 2023 would be corresponding to 'year' = 2023 and 'day_of_year' = 10.

Let's filter the payments data to find transactions for Belles_cookbook_store on the 10th day of 2023, and then identify which Fee IDs would apply to those transactions based on the fees.json file and the merchant characteristics.

Code:
```py
import pandas as pd
import json

# Read the necessary files
with open('data/context/merchant_data.json', 'r') as file:
    merchant_data = json.load(file)

with open('data/context/fees.json', 'r') as file:
    fees_data = json.load(file)

payments_df = pd.read_csv('data/context/payments.csv')

# Find Belles_cookbook_store data
belles_data = None
for merchant in merchant_data:
    if merchant.get('merchant') == 'Belles_cookbook_store':
        belles_data = merchant
        break

if belles_data:
    print("Belles_cookbook_store characteristics:")
    print(f"Account Type: {belles_data['account_type']}")
    print(f"Merchant Category Code: {belles_data['merchant_category_code']}")
    print(f"Capture Delay: {belles_data['capture_delay']}")

    # Filter payments for Belles_cookbook_store on day 10 of 2023
    belles_payments_day10 = payments_df[
        (payments_df['merchant'] == 'Belles_cookbook_store') &
        (payments_df['year'] == 2023) &
        (payments_df['day_of_year'] == 10)
    ]
```

*Continued*

Figure 8: Agent step 6, part 0, where agent is retrieving payment traffic from the merchant. The full code generated by the agent in step 6 is continued in 9.

**Agent**

```
   if not belles_payments_day10.empty:
       print(f"\nFound {len(belles_payments_day10)} payments for Belles_cookbook_store on day 10
of 2023")

       # Get unique combinations of relevant fields for checking fee applicability
       unique_txn_types = belles_payments_day10[['card_scheme', 'is_credit', 'aci']].drop_duplicates()
       print("\nUnique transaction types:")
       print(unique_txn_types)

       # Find applicable fee IDs
       applicable_fees = []

       for _, txn_type in unique_txn_types.iterrows():
          card_scheme = txn_type['card_scheme']
          is_credit = txn_type['is_credit']
          aci = txn_type['aci']

          for fee in fees_data:
              # Check if fee applies to this merchant and transaction type
              scheme_match = fee['card_scheme'] == card_scheme or fee['card_scheme'] is None
              account_type_match = not fee['account_type'] or belles_data['account_type'] in
fee['account_type']
              mcc_match = not fee['merchant_category_code'] or belles_data['merchant_category_code']
in fee['merchant_category_code']
              credit_match = fee['is_credit'] == is_credit or fee['is_credit'] is None
              aci_match = not fee['aci'] or aci in fee['aci']

              if scheme_match and account_type_match and mcc_match and credit_match and aci_match:
                 if fee['ID'] not in applicable_fees:
                     applicable_fees.append(fee['ID'])

       applicable_fees.sort()
       print("\nApplicable Fee IDs:")
       print(applicable_fees)
   else:
       print("\nNo payments found for Belles_cookbook_store on day 10 of 2023")
else:
   print("Belles_cookbook_store not found in merchant_data.json")
```
```

Figure 9: Agent step 6, part 1, where the agent joins the payment traffic information with the rules about fee calculations it has learned about. Step 6 is continued in 10.

**Step 6**

**Execution Logs**

Belles_cookbook_store characteristics:
Account Type: R
Merchant Category Code: 5942
Capture Delay: 1

Found 37 payments for Belles_cookbook_store on day 10 of 2023

Unique transaction types:
```
       card_scheme  is_credit aci
6042        NexPay     False  F
7109      GlobalCard   False  A
7961      GlobalCard    True  D
10061     GlobalCard    True  G
23853       NexPay      True  D
35807   TransactPlus    True  G
36239   TransactPlus    True  A
45238     GlobalCard    True  C
58584     GlobalCard   False  F
64920     GlobalCard   False  D
78788   TransactPlus    True  F
79891   TransactPlus    True  D
96367       NexPay      True  F
98810     SwiftCharge  False  F
108163    GlobalCard    True  F
111835      NexPay     False  G
131861  TransactPlus   False  F
```

Applicable Fee IDs:
[29, 78, 121, 286, 304, 357, 359, 381, 427, 454, 456, 473, 477, 484, 502, 536, 572, 596, 602, 606, 642, 704, 709, 741, 792, 813, 861, 871, 895, 917, 944, 968]

Figure 10: Agent step 6, part 2, where the execution logs show the found fees by the agent.

**Step 7**

**Agent**

Thought: Now I have the information I need to provide the answer. I've found the applicable Fee IDs for Belles_cookbook_store on the 10th day of the year 2023. According to the analysis, there are 32 fee IDs that apply:
[29, 78, 121, 286, 304, 357, 359, 381, 427, 454, 456, 473, 477, 484, 502, 536, 572, 596, 602, 606, 642, 704, 709, 741, 792, 813, 861, 871, 895, 917, 944, 968]

I should format them according to the guidelines provided: "Answer must be a list of values in comma separated list, eg: A, B, C."

Code:
```py
# Format the answer as a comma-separated list
fee_ids = [29, 78, 121, 286, 304, 357, 359, 381, 427, 454, 456, 473, 477, 484, 502, 536, 572, 596, 602, 606, 642, 704, 709, 741, 792, 813, 861, 871, 895, 917, 944, 968]

# Convert integers to strings and join with commas
formatted_answer = ", ".join(str(fee_id) for fee_id in fee_ids)

# Provide the final answer
final_answer(formatted_answer)
```

**Execution Logs**

Final answer: 29, 78, 121, 286, 304, 357, 359, 381, 427, 454, 456, 473, 477, 484, 502, 536, 572, 596, 602, 606, 642, 704, 709, 741, 792, 813, 861, 871, 895, 917, 944, 968

Figure 11: Agent step 7 where agent produces a final answer compliant with the task guidelines.

