# OpenReview forum: "DABstep: Data Agent Benchmark for Multi-step Reasoning"
_NeurIPS.cc/2025/Datasets_and_Benchmarks_Track — Submitted to NeurIPS 2025 Datasets and Benchmarks Track_

### Official Review · Reviewer_dNfK · 2025-06-11

**Rating:** 5
**Confidence:** 4

**Summary:**

This paper introduces DABstep, which benchmarks AI Agents in multi-step data analysis tasks. The benchmark comprises 450 samples derived from 95 “core” questions. The authors benchmark ~16 models, both instruction-tuned and reasoning models, of varying capabilities, showing that AI agents excel at “easy” (1 shot analysis) tasks, but struggle in more demanding analysis settings. The authors conclude with a high-level discussion about agent failure modes.

**Additional Feedback:**

I have a few additional questions about the evaluation / benchmark:
- Did the authors perform any validation to ensure that the factoid-based answers are unique?
- Did the authors validate that each task is solvable within 10 steps? The evaluation setup limits agents to 10 ReAct steps.
- Nit: It’d be great to have some more figures in the paper, perhaps presenting finer-grained analysis in the form of easy-to-read charts and plots. The current paper is very text-dense.

**Dataset Code Accessibility:**

Yes

**Dataset Code Comments:**

The authors released code and the test-set questions publicly. A Hugging Face page is provided for users to submit their model outputs for evaluation.

**Ethical Considerations:**

No, there are no or only very minor ethics concerns

**Final Justification:**

My main questions/concerns about finer-grained analysis were thoroughly addressed by the authors with their new taxonomy. I believe its addition will make the paper stronger, and have updated my score to reflect this. I believe this benchmark will be a valuable contribution.

**Limitations Weaknesses:**

The paper presents a valuable benchmark, but there are a few limitations in its current form. I’m looking forward to hearing the authors’ responses.
- **Limited fine-grained analysis**: The authors highlight four observed failure modes in Section 4.3, e.g., models write inefficient code, and provide an example trace in Appendix A.4. However, the analysis in Section 4.3 is only describing global trends. As a reader, I would love to learn more about fine-grained analysis, which given the hidden ground-truth labels, only the authors can perform. Some questions I have:
  - Is there a finer-grained categorization of the 95 core questions? If so, do models exhibit better or worse performance for specific core question types? How do these trends hold over different model families?
  - Given the GSM-Symbolic construction of the test set, are there any interesting trends about how agents handle different variations of the same core question? What is the “pass-rate” of each core question for the models? How do the best/worst models handle parametrizations?
  - How far along do agents get within their allotted 10 ReAct steps? Do agents make meaningful progress or do they get stuck in loops? Do stronger models make more meaningful progress than weaker models?
- **Relatively small test-set requires statistical significance testing**: DABstep appears to be on the smaller size in terms of test-set size (450 samples) when compared to existing related benchmarks (Table 2), and can be considered even smaller given that samples are parametrized versions of 95 core questions. I think some statistical significance testing should be done to validate if the top performing models are statistically different in their performance or not.
- **Focus on single-agent systems**: Multi-agent systems have become more popular recently, e.g., ADAS [1], AFLOW [2], etc. Did the authors try testing the efficacy of such systems? One drawback of some multi-agent systems is that they require a validation set in order to determine an agentic design for test-time. Given the parametric setup of DABstep, perhaps said systems could be tuned on a separately generated validation set? This analysis/benchmarking would be nice, but is optional given higher costs / data generation needed.

[1] https://arxiv.org/abs/2408.08435
[2] https://arxiv.org/abs/2410.10762

**Strengths Contributions:**

- The authors are addressing an important setting for AI agents, which to my knowledge, has not been covered in-depth prior. The proposed DABstep construction and evaluation methodology seem sound.
- In particular, I appreciate that the authors kept test-set solutions hidden and standardized the evaluation protocol, which should hopefully guarantee contamination-free evaluation going forward. I also appreciate the focus of ease-of-use and ease-of-submission.
- The authors’ evaluation of single-agent systems is fairly comprehensive, testing many different models of different capabilities.
- The paper is well-written and easy to follow, clearly covering related work (Table 2) and outlining dataset construction and evaluation methodology.

---

> ### Author Rebuttal · Authors · 2025-07-29
>
> # Rebuttal to Reviewer dNfK
>
> Thank you for your detailed and thoughtful review. We’re glad you found the benchmark’s design, clarity, and evaluation sound. Below we address your main concerns:
>
> ---
>
> ### Fine-Grained Performance Analysis on Core Questions
>
> We agree that a more granular breakdown of task complexity is a significant improvement. To address this, we have developed and are integrating a **multi-dimensional complexity rubric** across all 95 core questions (**see Table 1 below**). This allows for much richer analysis of agent performance and directly supports the finer-grained evaluation you suggested.
>
> The rubric scores each task across five dimensions:
>
> 1. **Cross-referenced data sources**
> 2. **Domain-specific transformations** (e.g., parsing MCC codes, fee aggregations)
> 3. **General-purpose transformations** (e.g., sorting, grouping, arithmetic)
> 4. **Explicit domain concepts** (stated in documentation)
> 5. **Implicit domain concepts** (unstated reasoning or linking requirements)
>
> We also compute a **normalized complexity score** by aggregating and normalizing across these axes. This lets us move beyond the binary Easy/Hard classification and understand performance as a function of nuanced task attributes.
>
> Considering Table 1:
> - **Tasks 1–72**: Low complexity across all axes
> - **Tasks 73–80**: Moderate complexity and generalization needs
> - **Tasks 81–95**: High complexity, especially in latent reasoning and domain alignment
> - **Tasks 93 & 95**: Highest overall complexity based on the rubric
>
> This framework also helps explain the modest 49% correlation between "Easy" and "Hard" task performance. It shows that "Hard" tasks introduce *new dimensions of difficulty*—not just longer versions of Easy tasks. We will include bar/radar charts in the camera-ready version to visualize this task-level complexity spectrum.
>
> ---
>
> ### Analysis of Agent Progress Within 10 Steps
>
> We appreciate the reviewer’s valid concern about whether the baselines actually get to meaningful answers within 10 steps. While an extended study on the effect of varying the “max steps” would give the reader a scientifically backed reason for our choice, we found empirically that a 10-step limit typically led either to a meaningful result or a clear failure mode. (NB: please don’t conflate agent steps with transformation steps, an agent can do multiple transformation per step)
>
> As mentioned wrt “Error Bars”, similar to doing a statistical analysis for confidence interval estimations, trying out all different configurations for “max steps” would be impractical, as it would be both computationally and financially infeasible (expensive).
>
> ---
>
> ### Statistical Significance Testing
>
> We appreciate this point. While DABStep includes 450 tasks, this is *not unusually small* by benchmark standards:
>
> | Benchmark      | # Tasks |
> |----------------|---------|
> | GAIA           | 466     |
> | Spider 2       | 632     |
> | KaggleDBQA     | 272     |
> | HumanEval      | 164     |
> | MLAgentBench   | 13      |
> | **DABStep**     | **450** |
>
> Moreover, **each DABStep task is deeply diagnostic**, involving multi-step planning, tool use, code execution, and document synthesis. As GSM‑Symbolic (arXiv:2410.05229) shows, even small variations over compact task sets can surface brittleness and test generalization.
>
> That said, we agree that statistical rigor is important. Repeated trials and significance testing are infeasible in terms of monetary expense given DABStep’s complexity, but we are prioritizing **trace-based evaluation** and **fine-grained scoring** in the upcoming leaderboard update—these will yield more robust diagnostics than aggregate accuracy metrics alone.
>
> We are also working toward a **1,000+ task expansion** in the next version to further strengthen the statistical foundation.
>
> ---
>
> ### Multi-Agent Evaluation
>
> This is a great suggestion. We did not evaluate multi-agent systems in this release primarily because **DABStep does not include a full development set**. Doing so would risk **leakage**, reducing fidelity for generalization benchmarking.
>
> However, we recognize that multi-agent systems—and tuning more broadly—are important for future work. We are considering to release a **separate full dev set** in future versions to enable safe tuning of such systems without compromising evaluation integrity.
>
> ---
>
> ### Unique Factoid Answers
>
> Yes—we confirm that each DABStep task answer is unique by design and will highlight this in the camera read version
>
> ---
>
> # Table 1 – Task Breakdown Summary
>
> We categorize each task along five key dimensions (table abbreviated for space):
>
> - **Cross References**: Number of distinct data sources (e.g., CSVs, JSONs, manuals).
> - **Domain Transformations**: Financial-specific data manipulations (e.g., fee parsing, fraud aggregation).
> - **General Transformations**: Standard operations (e.g., grouping, date parsing, arithmetic).
> - **Explicit Concepts**: Clearly defined domain rules from documentation.
> - **Implicit Concepts**: Latent or inferred relationships requiring contextual reasoning (see Section 4.3).
>
> *Note: Transformation counts refer to conceptual steps, not one-to-one agent actions.*
>
>
> | ID | L | XR | D | G | E | I | S |
> |----|---|----|---|---|---|---|----|
> | 1-72 | E | 2.5 | 0 | 1.5 | 0.5 | 0 | 0.167 |
> | 73 | H | 2 | 0 | 3 | 0 | 0 | 0.186 |
> | 74 | H | 2 | 2 | 2 | 1 | 1 | 0.297 |
> | 75 | H | 3 | 3 | 2 | 1 | 2 | 0.408 |
> | 76 | H | 2 | 2 | 4 | 1 | 1 | 0.371 |
> | 77 | H | 2 | 3 | 5 | 1 | 2 | 0.483 |
> | 78 | H | 2 | 2 | 1 | 0 | 2 | 0.260 |
> | 79 | H | 2 | 2 | 2 | 1 | 0 | 0.260 |
> | 80 | H | 2 | 2 | 3 | 1 | 1 | 0.334 |
> | 81 | H | 5 | 13 | 6 | 3 | 8 | 1.299 |
> | 82 | H | 5 | 14 | 6 | 4 | 8 | 1.373 |
> | 83 | H | 5 | 13 | 6 | 3 | 8 | 1.299 |
> | 84 | H | 5 | 14 | 6 | 4 | 8 | 1.373 |
> | 85 | H | 5 | 13 | 6 | 3 | 8 | 1.299 |
> | 86 | H | 5 | 14 | 6 | 4 | 8 | 1.373 |
> | 87 | H | 5 | 15 | 8 | 4 | 8 | 1.485 |
> | 88 | H | 5 | 15 | 8 | 4 | 8 | 1.485 |
> | 89 | H | 5 | 14 | 7 | 3 | 8 | 1.373 |
> | 90 | H | 5 | 15 | 8 | 4 | 8 | 1.485 |
> | 91 | H | 5 | 15 | 8 | 3 | 8 | 1.448 |
> | 92 | H | 5 | 15 | 7 | 4 | 8 | 1.448 |
> | 93 | H | 5 | 15 | 7 | 4 | 10 | 1.522 |
> | 94 | H | 5 | 15 | 7 | 4 | 8 | 1.448 |
> | 95 | H | 5 | 15 | 7 | 4 | 10 | 1.522 |
> | **TOTALS** |   | **272** | **231** | **233** | **97** | **133** |      |
>
> Key:
>
> * `L`: Legacy (`E` = Easy, `H` = Hard)
> * `XR`: Cross References
> * `D`: Transformations (domain)
> * `G`: Transformations (general)
> * `E`: Domain Concepts (explicit)
> * `I`: Domain Concepts (implicit)
> * `S`: Normalized Score

---

> > ### Author Response · Authors · 2025-08-05
> >
> > Dear Reviewer,
> >
> > We've posted our rebuttal addressing your comments. Would you be available to discuss any remaining questions or concerns?
> >
> > Thank you for your time.

---

> > ### Comment · Reviewer_dNfK · 2025-08-05
> > **Response to Authors**
> >
> > Thank you for your thorough responses. I think the task categorization/rubric and related analysis will make this paper much stronger when included in the updated manuscript. I will update my rating accordingly.

---

### Official Review · Reviewer_wMuq · 2025-07-01

**Rating:** 4
**Confidence:** 2

**Summary:**

DABstep introduces a valuable benchmark for evaluating AI agents on realistic multi-step data analysis tasks, demonstrating significant performance gaps in current state-of-the-art models with the best agent achieving only 14.55% accuracy on hard tasks. While the benchmark successfully establishes objective evaluation standards and highlights critical limitations in agent reasoning capabilities, it suffers from limited domain coverage, oversimplified task classification, and insufficient failure mode analysis. The work provides a solid foundation for future research in autonomous data analysis, but requires expansion beyond financial domains and more comprehensive evaluation methodologies to achieve its claimed generalizability. Despite these limitations, DABstep fills an important gap in agent evaluation and offers clear directions for improving multi-step reasoning capabilities in AI systems.

**Dataset Code Accessibility:**

Yes

**Dataset Code Comments:**

The authors provide comprehensive accessibility through HuggingFace platforms with all materials readily available. The benchmark dataset is hosted at https://huggingface.co/datasets/adyen/dabstep with detailed dataset cards, while the complete codebase including baseline implementations and evaluation scripts is available at https://huggingface.co/spaces/adyen/DABstep/tree/main/baseline.

**Ethical Considerations:**

No, there are no or only very minor ethics concerns

**Final Justification:**

This paper presents a valuable benchmark that addresses an important gap in evaluating AI agents on realistic multi-step data analysis tasks. The work demonstrates strong technical merit through its comprehensive evaluation methodology, well-documented codebase, and commitment to reproducibility via the public leaderboard and HuggingFace hosting. The authors' development of a five-dimensional complexity rubric in response to reviewer feedback significantly strengthens the contribution by providing granular analysis beyond the initial binary Easy/Hard classification. The benchmark successfully reveals substantial performance gaps in current state-of-the-art models, with even the best agent achieving only 14.55% accuracy on the hardest tasks, highlighting critical areas for future research in autonomous data analysis.


While I maintain concerns about the benchmark's scale limitation with only 450 tasks derived from 95 core questions, I acknowledge that the task depth and diagnostic value partially compensate for this constraint. The comparison table provided by the authors includes benchmarks with varying complexity profiles, and citing smaller datasets like HumanEval to justify scale may not fully address the statistical reliability concerns for comprehensive evaluation across multiple complexity dimensions. However, the authors' commitment to expanding to 1,000+ tasks in future versions and their plans for multi-domain expansion beyond financial analytics demonstrate awareness of these limitations and a clear path forward. The benchmark's focus on factoid-style evaluation, while enabling objective assessment, represents a deliberate trade-off that may limit assessment of complete analytical workflows, but this choice ensures scalable and unbiased evaluation that will benefit the research community.

**Limitations Weaknesses:**

1. The 450 tasks are relatively small compared to mainstream benchmarks (e.g., MATH: 12,500, Spider: 80,654). The binary Easy/Hard classification is oversimplified and should include fine-grained skill dimensions like data manipulation, statistical analysis, and cross-source reasoning. The benchmark lacks quantitative complexity metrics and distribution statistics for required steps and data sources.
2. Task descriptions on HuggingFace are overly brief, lacking sufficient context to activate LLM reasoning capabilities. The benchmark uses only standardized ReAct prompts without optimization for multi-step reasoning tasks. Missing few-shot examples and detailed reasoning guidance may inadequately challenge the models' true reasoning potential.
3. The paper lacks quantitative analysis of failure mode frequency distribution and severity classification. There is insufficient root cause analysis distinguishing technical failures (code errors) vs. reasoning failures (logical errors). The authors cannot identify which failure modes are most critical or which are fundamentally unsolvable by LLM improvements alone.
4. The title "Data Agent Benchmark" implies general data analysis capability but the content is highly specialized in financial payments. Tasks heavily rely on domain-specific knowledge (MCC codes, ACI features, fee structures), potentially testing domain expertise rather than pure reasoning ability. This creates accessibility barriers for non-finance researchers and misaligns with the claimed generalizability.
5. Single-run evaluation without confidence intervals or statistical significance testing ignores model output randomness. The binary accuracy metric is oversimplified and lacks partial correctness assessment and reasoning process quality evaluation. The benchmark should include intermediate step correctness evaluation rather than focusing solely on final answers.

**Strengths Contributions:**

1. The benchmark provides 450 authentic tasks derived from actual operational workloads at Adyen, ensuring real-world relevance and complexity. The factoid-based evaluation framework with hybrid scoring algorithm effectively avoids LLM-as-a-judge biases while maintaining objectivity and scalability.
2. Tasks are explicitly designed to require multi-step reasoning with iterative data processing across heterogeneous sources, addressing a critical gap in existing benchmarks. The low-barrier setup requiring only standard Python runtime and automated leaderboard facilitates broad community participation and reproducible evaluation.
3. The paper is clearly written with logical flow and comprehensive methodology description, making it easy to follow and understand. The authors provide complete open-source implementation including datasets, baseline code, evaluation scripts, and public leaderboard on HuggingFace, demonstrating strong commitment to reproducibility and community engagement.

---

> ### Author Rebuttal · Authors · 2025-07-29
>
> # Rebuttal to Reviewer wMuq
>
> Thank you for your constructive review. We appreciate your recognition of the benchmark’s realism, accessibility, and multi-step complexity. We address your key concerns below:
>
> ---
>
> ### Benchmark Size Compared to Other Datasets
>
> We acknowledge the observation regarding DABStep's scale. While our benchmark currently includes 450 tasks—fewer than some large-scale datasets—we emphasize that **evaluation quality cannot be measured by task count alone**.
>
> Below, we contextualize DABStep’s scale relative to other well-regarded benchmarks. As shown, many impactful datasets fall within a similar range or smaller:
>
> | Rank | Benchmark      | # Tasks |
> |------|----------------|---------|
> | 1    | WikiSQL        | 80,654  |
> | 2    | BIRD           | 12,751  |
> | 3    | MATH           | 12,500  |
> | …    | …              | …       |
> | 14 | Spider 2       |       632 |
> | 15 | DSBench        |       540 |
> | 16 | DA-Code        |       500 |
> | 17 | Spider2-V      |       494 |
> | 18   | GAIA           | 466     |
> | 19   | **DABStep**     | **450** |
> | 20   | OSWorld        | 369     |
> | 21   | KaggleDBQA     | 272     |
> | 22   | DABench        | 257     |
> | 23   | HumanEval      | 164     |
> | 24   | MLAgentBench   | 13      |
>
> As shown, many highly regarded benchmarks (e.g., GAIA, HumanEval, Spider) are similarly sized or smaller. What distinguishes DABStep is **task depth**: each task requires multi-step reasoning, code generation, document integration, and tool-based planning grounded in real-world analytic workflows.
>
> Our **parameterization strategy** is not intended to inflate volume artificially, but rather to create *structured variations* that test generalization and consistency, in line with methodologies like *GSM-Symbolic* (arXiv:2410.05229). We are currently working toward an expansion to 1,000+ tasks in future releases while maintaining high diagnostic value.
>
> ---
>
> ### Binary Easy/Hard Classification
>
> We fully agree that binary task labeling oversimplifies complexity. To address this, we have developed and integrated a _multi-dimensional task complexity rubric_ for all 95 core questions, capturing five difficulty dimensions (**see Table 1 below**):
>
> 1. **Cross-Referenced Sources**
> 2. **General Transformations** (e.g., sorting, grouping, date/time ops)
> 3. **Domain-Specific Transformations** (e.g., parsing fees or MCC codes)
> 4. **Explicit Domain Concepts** (stated in documentation)
> 5. **Implicit Domain Concepts** (unstated dependencies or logic)
>
> We compute a **normalized complexity score** by aggregating these components. This reveals a clear complexity gradient across tasks—notably, tasks 81–95 require significantly more domain reasoning and multi-source alignment than tasks 1–72 (** see Table 1 below**).
>
> We hope this analysis begins to addresses the reviewer’s call for fine-grained metrics and confirms that task difficulty stems from **reasoning complexity—not just domain familiarity**.
>
> ---
>
> ### Brief Task Descriptions on HuggingFace
>
> Thank you for this feedback. We will revise the HuggingFace task descriptions to include **richer context** and clearer task objectives to support both LLM reasoning and developer use.
>
> ---
>
> ### Prompting Limitations
>
> Our prompting strategy—standardized few-shot ReAct-style prompts—was designed to ensure **fairness and reproducibility** across model classes. As discussed in Sections 4.1 and 4.3, we acknowledge this may underestimate peak model performance and frame our results as **lower bounds**.
>
> For known prompt-sensitive models (e.g., o4-mini, Deepseek-R1), we applied modest prompt tuning to ensure fair comparisons.
>
> We view the live and open leaderboard as a venue for broader prompt experimentation beyond our baseline setup.
>
> ---
>
> ### Lack of Intermediate-Step Evaluation
>
> We appreciate this thoughtful point. Our current end-to-end evaluation prioritizes **objectivity and automation** through factoid-style answers. This avoids recursive issues posed by LLM-as-a-judge approaches, which we discuss critically in the paper.
>
> That said, we **fully agree** on the value of trace-based evaluation. We are actively developing support for:
>
> - **Intermediate step scoring**
> - **Partial credit**
> - **Failure type categorization**
> - **Trajectory logging for reasoning traces**
>
> These capabilities are planned for upcoming leaderboard updates and will enable more nuanced analysis of agent reasoning.
>
> Regarding significance testing: repeated runs across hundreds of tasks is prohibitively expensive at present, but our future trace-based scoring should offer **more scalable diagnostics** of robustness and failure patterns.
>
> ---
>
> ### Domain Specialization and Accessibility
>
> We appreciate the concern about domain specificity. The financial domain was selected due to access to **authentic, well-instrumented tasks** from Adyen with high-quality ground truth. This ensures realism and relevance to real-world multi-step analytics.
>
> To assess generalizability, we analyzed all tasks using the a new complexity rubric (**see Table 1 below**). The key findings:
>
> - **General-purpose transformations** (e.g., arithmetic, filtering): **233**
> - **Domain-specific transformations**: **231**
> - Many "hard" tasks test *implicit logic*, not financial knowledge
>
> This shows that **general reasoning is as central as domain expertise**. Tasks 74–80, for example, contain just 2 domain-specific steps, while requiring general-purpose planning and document synthesis.
>
> DABStep is designed to test **agents**, not humans. The expectation is not domain mastery, but whether an agent can **adapt and reason** through unfamiliar yet structured problems. This mirrors realistic challenges faced by agents in varied industries.
>
> We are expanding DABStep to **healthcare, e-commerce, and logistics** (see Section 5), and are working to **annotate tasks** with clearer metadata and create **domain-agnostic baselines** to further reduce barriers.
>
> Ultimately, the name *"Data Agent Benchmark"* reflects our goal: to test and accelerate **autonomous agents capable of real-world, multi-step data analysis**—starting with one grounded domain and scaling outward responsibly.
>
>
> ---
>
> # Table 1 – Task Breakdown Summary
>
> We categorize each task along five key dimensions (table abbreviated for space):
>
> - **Cross References**: Number of distinct data sources (e.g., CSVs, JSONs, manuals).
> - **Domain Transformations**: Financial-specific data manipulations (e.g., fee parsing, fraud aggregation).
> - **General Transformations**: Standard operations (e.g., grouping, date parsing, arithmetic).
> - **Explicit Concepts**: Clearly defined domain rules from documentation.
> - **Implicit Concepts**: Latent or inferred relationships requiring contextual reasoning (see Section 4.3).
>
> *Note: Transformation counts refer to conceptual steps, not one-to-one agent actions.*
>
>
> | ID | L | XR | D | G | E | I | S |
> |----|---|----|---|---|---|---|----|
> | 1-72 | E | 2.5 | 0 | 1.5 | 0.5 | 0 | 0.167 |
> | 73 | H | 2 | 0 | 3 | 0 | 0 | 0.186 |
> | 74 | H | 2 | 2 | 2 | 1 | 1 | 0.297 |
> | 75 | H | 3 | 3 | 2 | 1 | 2 | 0.408 |
> | 76 | H | 2 | 2 | 4 | 1 | 1 | 0.371 |
> | 77 | H | 2 | 3 | 5 | 1 | 2 | 0.483 |
> | 78 | H | 2 | 2 | 1 | 0 | 2 | 0.260 |
> | 79 | H | 2 | 2 | 2 | 1 | 0 | 0.260 |
> | 80 | H | 2 | 2 | 3 | 1 | 1 | 0.334 |
> | 81 | H | 5 | 13 | 6 | 3 | 8 | 1.299 |
> | 82 | H | 5 | 14 | 6 | 4 | 8 | 1.373 |
> | 83 | H | 5 | 13 | 6 | 3 | 8 | 1.299 |
> | 84 | H | 5 | 14 | 6 | 4 | 8 | 1.373 |
> | 85 | H | 5 | 13 | 6 | 3 | 8 | 1.299 |
> | 86 | H | 5 | 14 | 6 | 4 | 8 | 1.373 |
> | 87 | H | 5 | 15 | 8 | 4 | 8 | 1.485 |
> | 88 | H | 5 | 15 | 8 | 4 | 8 | 1.485 |
> | 89 | H | 5 | 14 | 7 | 3 | 8 | 1.373 |
> | 90 | H | 5 | 15 | 8 | 4 | 8 | 1.485 |
> | 91 | H | 5 | 15 | 8 | 3 | 8 | 1.448 |
> | 92 | H | 5 | 15 | 7 | 4 | 8 | 1.448 |
> | 93 | H | 5 | 15 | 7 | 4 | 10 | 1.522 |
> | 94 | H | 5 | 15 | 7 | 4 | 8 | 1.448 |
> | 95 | H | 5 | 15 | 7 | 4 | 10 | 1.522 |
> | **TOTALS** |   | **272** | **231** | **233** | **97** | **133** |      |
>
> Key:
>
> * `L`: Legacy (`E` = Easy, `H` = Hard)
> * `XR`: Cross References
> * `D`: Transformations (domain)
> * `G`: Transformations (general)
> * `E`: Domain Concepts (explicit)
> * `I`: Domain Concepts (implicit)
> * `S`: Normalized Score

---

> > ### Author Response · Authors · 2025-08-05
> >
> > Dear Reviewer,
> >
> > We've posted our rebuttal addressing your comments. Would you be available to discuss any remaining questions or concerns?
> >
> > Thank you for your time.

---

> > > ### Comment · Reviewer_wMuq · 2025-08-05
> > > **Concern about the scale**
> > >
> > > Thank you for the comprehensive and thoughtful responses. I appreciate the significant effort put into addressing the concerns, particularly the development of the five-dimensional complexity rubric and the detailed analysis of domain-specific versus general transformations.
> > >
> > > However, I remain unconvinced regarding the benchmark scale limitation. The comparison table includes benchmarks with vastly different complexity profiles, and it's unclear what criteria were used to generate this ranking or selection. Citing simple datasets like HumanEval (164 tasks) to justify DABstep's scale misses the point  that **complex, comprehensive benchmarks inherently require larger scales for robust evaluation**.
> > >
> > > With only 450 tasks derived from 95 core questions, the benchmark risks statistical unreliability  across multiple complexity dimensions, potentially limiting the generalizability of findings and increasing susceptibility to overfitting specific task formulations.
> > >
> > > I maintain my borderline accept rating, acknowledging the valuable contribution while recognizing the scale limitation as a constraint on the work's immediate impact.

---

### Official Review · Reviewer_9HH8 · 2025-07-02

**Rating:** 4
**Confidence:** 3

**Summary:**

he authors propose DABstep, a benchmark for evaluating AI agents on realistic multi-step data analysis tasks. The benchmark comprises over 450 real-world challenges derived from a financial analytics platform, requiring models to combine code-based data processing with contextual reasoning over heterogeneous documentation (structured data like CSV/JSON and unstructured text like Markdown manuals). Each task demands iterative, multi-step problem-solving approaches, testing capabilities in data manipulation, cross-referencing multiple sources, and precise result reporting. The benchmark uses factoid-style answers with automatic correctness checks for objective scoring. Baseline evaluations reveal substantial performance gaps, with the best agent (o4-mini) achieving only 14.55% accuracy on the hardest tasks. The authors release the benchmark with a public leaderboard and toolkit to accelerate research in autonomous data analysis.

**Additional Feedback:**

The benchmark would benefit from more granular difficulty metrics beyond binary Easy/Hard classification and quantitative error type distributions across models and task categories.

The 49% correlation between Easy and Hard tasks suggests they may test different capabilities requiring deeper investigation.

The prompt sensitivity observed for reasoning models indicates the standardized evaluation approach may not capture true model capabilities.

Figure 1 could benefit from clearer visual differentiation between example tasks.

The trace example in the appendix could be summarized more effectively in the main text. I found it important to understanding the contribution.

**Dataset Code Accessibility:**

Yes

**Dataset Code Comments:**

The authors provide access through HuggingFace hosting, detailed baseline implementations, pseudocode algorithms, and developer tools. The CC-BY-4.0 licensing ensures broad usability.

**Ethical Considerations:**

No, there are no or only very minor ethics concerns

**Final Justification:**

I still have some concerns on the claim of the results being domain transferable.  I will keep my score.

**Limitations Weaknesses:**

The exclusive focus on financial analytics limits generalizability, as domain-specific knowledge requirements may not transfer to other analytical domains, potentially restricting the benchmark's broader applicability for evaluating general data analysis capabilities.

The deliberate choice to focus on factoid answers, while enabling objective evaluation, excludes important real-world analytical tasks requiring narrative summaries, recommendations, or exploratory analysis, representing a significant trade-off that may limit assessment of complete analytical workflows.

The expansion from 95 core questions to 450 tasks through systematic parameterization makes me question the true task diversity versus surface-level adaptability.

The standardized prompting approach may significantly underestimate model capabilities, particularly for reasoning models that performed poorly with ReAct prompts but improved with reasoning-specific prompts.

The binary Easy/Hard classification may oversimplify task complexity.

The failure mode analysis lacks quantitative breakdowns of error types, with specific examples relegated to appendix figures without systematic analysis of patterns across different models and task types.

**Strengths Contributions:**

DABstep addresses a critical gap in existing benchmarks by focusing explicitly on multi-step reasoning complexity, with 84% of tasks requiring iterative data filtering, cross-referencing, and multi-stage planning rather than single-shot generation.

The benchmark's derivation from actual workloads in financial analytics provides authentic challenges that integrate structured data (payments.csv with 100k+ transactions) and unstructured documentation, mirroring real-world scenarios where analysts must synthesize multiple heterogeneous information sources.

The factoid-based evaluation approach with a hybrid scoring algorithm provides a robust and objective assessment that avoids LLM-as-a-judge biases while handling semantic variations that showed a 100% agreement with human judgment on 75 examples.

The comprehensive evaluation of 16 state-of-the-art models reveals valuable insights into current limitations with systematic failure mode analysis identifying specific deficiencies in planning and instruction following.

---

> ### Author Rebuttal · Authors · 2025-07-29
>
> # Rebuttal to Reviewer 9HH8
>
> Thank you for your detailed review and insightful critique. We address your main concerns below:
>
> ---
>
> ### **Domain specificity (finance)**
>
> While the current benchmark focuses on financial analytics, the *skills required by the tasks—joins, filtering, aggregations, and cross-referencing structured and unstructured data—are general-purpose* and not domain-locked. The financial setting was chosen primarily for access to high-quality, real-world tasks from Adyen with clean ground truth. However, the core reasoning patterns and capabilities tested—tool-augmented multi-step planning, document integration, contextual retrieval, and intermediate result handling—are *domain-agnostic and transferable*.
>
> To support this with empirical evidence, and motivated by your gracious review, we have developed a **multi-dimensional complexity rubric** across all 95 core questions, decomposing each task into five categories (**see Table 1 below**):
>
> 1. Cross-referenced data sources
> 2. General-purpose data transformations
> 3. Domain-specific transformations
> 4. Explicit domain concepts
> 5. Implicit domain concepts
>
> Across the entire benchmark, **233 of the required transformation steps are general-purpose**, compared to **231 that are domain-specific**—demonstrating that general data manipulation skills are at least equally important. Moreover, many of the hardest tasks test implicit reasoning or document alignment, not finance-specific logic.
>
> This richer analysis demonstrates that *task difficulty arises from reasoning and planning challenges*, not just financial knowledge. It also provides a scalable foundation for tracking model capabilities across task types and underpins our longstanding plan to expand into diverse domains such as healthcare and e-commerce (see Section 5).
>
> ---
>
> ### **Factoid answer limitation**
>
> We recognize that our current framework focuses on factoid or verifiable tasks, which excludes more open-ended aspects of data analysis such as exploratory reasoning or strategic recommendations. This was a deliberate tradeoff: objective, scalable evaluation is only feasible when tasks have clear, verifiable answers.
>
> While some recent work explores *LLM-as-a-judge* approaches, we caution against this in the paper, as it introduces a recursive evaluation problem—how to evaluate both the agent’s answer and the judge itself. That said, we are actively exploring hybrid approaches for future releases, including **intermediate reasoning trace evaluation**, which would support partial credit and allow for more flexible task formulations.
>
> ---
>
> ### **True task diversity (parameterized tasks)**
>
> We appreciate the reviewer’s concern about whether our parameterization—from 95 to 450 tasks—reflects genuine task diversity or just surface-level changes. While we start from core questions, each parameterized instance is crafted to test meaningful generalization—such as shifts in dates, entities, or structural framing—rather than merely inflating volume.
>
> As noted in Section 3.1, this design aims to **reduce memorization and evaluate agent consistency**. We draw inspiration from *GSM-Symbolic* (arXiv:2410.05229), which similarly found that even minimal perturbations (e.g., swapping numbers or altering context phrasing) significantly impact LLM performance. Our structured variation is therefore intended to **stress-test brittle reasoning and expose generalization gaps**. We are also adding metadata to annotate variation types and core question families to improve transparency.
>
> ---
>
> ### **Prompting limitations**
>
> We agree that standardized prompting may understate best-case model performance and explicitly note this in Section 4.3. Our choice to standardize prompts across model classes was driven by a desire for fairness and reproducibility. These results should be interpreted as **lower bounds**, but they still offer valuable insights into *relative model capabilities*—especially in areas critical to practical data analysis (e.g., tool use, planning, instruction following).
>
> That said, we did apply **reasoning-optimized prompts** for certain models (e.g., o4-mini, o1). Our open and continuously updated leaderboard supports prompt variants, and we encourage the community to submit optimized runs.
>
> ---
>
> ### **Task complexity granularity & failure mode quantification**
>
> We agree that a more granular breakdown of task complexity provides deeper insight into model performance. In response, we developed a **five-dimensional complexity rubric** for all 95 core questions (**see Table 1 below**). The categories are:
>
> 1. Cross-referenced data sources
> 2. Domain-specific transformations
> 3. General-purpose transformations
> 4. Explicit domain concepts
> 5. Implicit domain concepts
>
> We also introduce a **normalized complexity score** per task, aggregating these dimensions.
>
> The totals across all tasks are:
> - **General transformations:** 233
> - **Domain transformations:** 231
> - **Explicit concepts:** 97
> - **Implicit concepts:** 133
>
> This shows that general reasoning is *at least as prominent* as domain-specific knowledge. Additionally, the sharp increase in implicit and multi-source reasoning in tasks 81–95 shows that 'Hard' tasks introduce *entirely new difficulty dimensions* rather than being scaled-up 'Easy' tasks.
>
> This framework also helps explain the modest 49% correlation between "Easy" and "Hard" task performance. It shows that "Hard" tasks introduce new dimensions of difficulty—not just longer versions of Easy tasks. We will include visualizations (e.g., radar and bar plots) of representative tasks in the final paper (unfortunately not embeddable here) to visualize this task-level complexity spectrum. We hope this deeper, principled breakdown directly addresses your request for finer-grained evaluation and significantly strengthens the paper’s contribution.
>
> ### Figure 1 Clarity & Appendix Trace
> We agree that Figure 1 can be improved to provide clearer visual differentiation among example tasks. In the camera-ready version, we will update Figure 1 by:
> * Introducing distinct visual markers or colors to differentiate task types or complexity levels clearly.
> * Labeling example tasks explicitly to illustrate the multi-dimensional complexity rubric.
> * Providing a brief annotation or caption describing key differences among tasks to enhance readability.
>
> We also agree that appendix trace would benefit the paper in the main text and will move to do this in the final camera ready version.
>
> ---
>
> # Table 1 – Task Breakdown Summary
>
> We categorize each task along five key dimensions (table abbreviated for space):
>
> - **Cross References**: Number of distinct data sources (e.g., CSVs, JSONs, manuals).
> - **Domain Transformations**: Financial-specific data manipulations (e.g., fee parsing, fraud aggregation).
> - **General Transformations**: Standard operations (e.g., grouping, date parsing, arithmetic).
> - **Explicit Concepts**: Clearly defined domain rules from documentation.
> - **Implicit Concepts**: Latent or inferred relationships requiring contextual reasoning (see Section 4.3).
>
> *Note: Transformation counts refer to conceptual steps, not one-to-one agent actions.*
>
>
> | ID | L | XR | D | G | E | I | S |
> |----|---|----|---|---|---|---|----|
> | 1-72 | E | 2.5 | 0 | 1.5 | 0.5 | 0 | 0.167 |
> | 73 | H | 2 | 0 | 3 | 0 | 0 | 0.186 |
> | 74 | H | 2 | 2 | 2 | 1 | 1 | 0.297 |
> | 75 | H | 3 | 3 | 2 | 1 | 2 | 0.408 |
> | 76 | H | 2 | 2 | 4 | 1 | 1 | 0.371 |
> | 77 | H | 2 | 3 | 5 | 1 | 2 | 0.483 |
> | 78 | H | 2 | 2 | 1 | 0 | 2 | 0.260 |
> | 79 | H | 2 | 2 | 2 | 1 | 0 | 0.260 |
> | 80 | H | 2 | 2 | 3 | 1 | 1 | 0.334 |
> | 81 | H | 5 | 13 | 6 | 3 | 8 | 1.299 |
> | 82 | H | 5 | 14 | 6 | 4 | 8 | 1.373 |
> | 83 | H | 5 | 13 | 6 | 3 | 8 | 1.299 |
> | 84 | H | 5 | 14 | 6 | 4 | 8 | 1.373 |
> | 85 | H | 5 | 13 | 6 | 3 | 8 | 1.299 |
> | 86 | H | 5 | 14 | 6 | 4 | 8 | 1.373 |
> | 87 | H | 5 | 15 | 8 | 4 | 8 | 1.485 |
> | 88 | H | 5 | 15 | 8 | 4 | 8 | 1.485 |
> | 89 | H | 5 | 14 | 7 | 3 | 8 | 1.373 |
> | 90 | H | 5 | 15 | 8 | 4 | 8 | 1.485 |
> | 91 | H | 5 | 15 | 8 | 3 | 8 | 1.448 |
> | 92 | H | 5 | 15 | 7 | 4 | 8 | 1.448 |
> | 93 | H | 5 | 15 | 7 | 4 | 10 | 1.522 |
> | 94 | H | 5 | 15 | 7 | 4 | 8 | 1.448 |
> | 95 | H | 5 | 15 | 7 | 4 | 10 | 1.522 |
> | **TOTALS** |   | **272** | **231** | **233** | **97** | **133** |      |
>
> Key:
>
> * `L`: Legacy (`E` = Easy, `H` = Hard)
> * `XR`: Cross References
> * `D`: Transformations (domain)
> * `G`: Transformations (general)
> * `E`: Domain Concepts (explicit)
> * `I`: Domain Concepts (implicit)
> * `S`: Normalized Score

---

> > ### Author Response · Authors · 2025-08-05
> >
> > Dear Reviewer,
> >
> > We've posted our rebuttal addressing your comments. Would you be available to discuss any remaining questions or concerns?
> >
> > Thank you for your time.

---

> > > ### Comment · Reviewer_9HH8 · 2025-08-08
> > >
> > > I understand your point on the tasks being relevant to multiple domains but I would argue that the claim that the results on the tasks are domain transferable should be empirically verified and is not guaranteed.  I am not sure how your response actually shows empirical evidence of the results across domains.

---

### Official Review · Reviewer_acdA · 2025-07-03

**Rating:** 5
**Confidence:** 4

**Summary:**

This paper introduces DABstep, a novel benchmark for evaluating AI agents on realistic multi-step data analysis tasks. This benchmark provides a challenging suite of data analysis agent evaluation tasks, making even the strongest LLM agents perform far from satisfactorily. This paper is excellently written, and the proposed dataset and analytical conclusions will be of great help to the community.

**Dataset Code Accessibility:**

Yes

**Dataset Code Comments:**

The dataset and code are well-documented, they are available at https://huggingface.co/spaces/adyen/DABstep.

**Ethical Considerations:**

No, there are no or only very minor ethics concerns

**Final Justification:**

Thank you for your responses, I look forward to multimodal capabilities. I think my score is high enough, so I will keep my score.

**Limitations Weaknesses:**

Although the provided benchmark is very useful, it does not yet support multimodal input.

**Strengths Contributions:**

Compared with other relevant benchmarks, the benchmark proposed in this paper more comprehensively evaluates the capabilities of LLMs.

The paper identifies limitations of best performing LLM agents and key areas for future research.

The benchmark is well-documented and easy to use.

A centralized live leaderboard to track progress and foster collaboration.

---

> ### Author Rebuttal · Authors · 2025-07-29
>
> Thank you for your encouraging and thoughtful review. We appreciate your positive remarks about the benchmark's scope, clarity, and potential impact on the community.
>
> Lack of multimodal input support:
> We agree that incorporating multimodal inputs (e.g., PDFs, charts, screenshots) is a critical direction for evaluating realistic data analysis workflows. As noted in Section 5, we plan to extend future versions of DABstep to include multimodal reasoning tasks. This initial release prioritized factoid-style questions to ensure objective, scalable evaluation and streamlined adoption, but multimodal capabilities are a top priority for upcoming iterations.
>
> We thank you again for recognizing the benchmark’s utility and community value.

---

> > ### Author Response · Authors · 2025-08-05
> >
> > Dear Reviewer,
> >
> > We've posted our rebuttal addressing your comments. Would you be available to discuss any remaining questions or concerns?
> >
> > Thank you for your time.

---

### Decision · Program_Chairs · 2025-09-18

**Decision:**

Reject

**Comment:**

The paper proposes DABStep, a benchmark for multi-step reasoning on data analysis of financial payments tasks. The benchmark is comprised of 450 tasks generated from 95 core tasks, and is challenging even for state-of-the-art models without scaffolding, leading to an accuracy of around 14%.
The benchmark focuses on the relatively unexplored area of multi-step reasoning for data analysis, as opposed to one-step generation, which was the focus of many previous benchmarks, and provides automatic verifiability via factoid-style answers.

While the focus on multi-step reasoning is timely, the focus on 95 core financial tasks makes it unclear in how far results on this dataset will generalized to other tasks and datasets. In particular, the benchmark contains only a single schema (Table 3), making it unsuitable for evaluating cross-schema generalization.